# P1 interneurons promote a persistent internal state that enhances inter-male aggression in *Drosophila*

Eric D Hoopfer[1,2], Yonil Jung[2], Hidehiko K Inagaki[2], Gerald M Rubin[1], David J Anderson[2,3]*

[1]Janelia Research Campus, Howard Hughes Medical Institute, Ashburn, United States; [2]Division of Biology and Biological Engineering, California Institute of Technology, Pasadena, United States; [3]Howard Hughes Medical Institute, California Institute of Technology, Pasadena, United States

**Abstract** How brains are hardwired to produce aggressive behavior, and how aggression circuits are related to those that mediate courtship, is not well understood. A large-scale screen for aggression-promoting neurons in *Drosophila* identified several independent hits that enhanced both inter-male aggression and courtship. Genetic intersections revealed that 8-10 P1 interneurons, previously thought to exclusively control male courtship, were sufficient to promote fighting. Optogenetic experiments indicated that P1 activation could promote aggression at a threshold below that required for wing extension. P1 activation in the absence of wing extension triggered persistent aggression via an internal state that could endure for minutes. High-frequency P1 activation promoted wing extension and suppressed aggression during photostimulation, whereas aggression resumed and wing extension was inhibited following photostimulation offset. Thus, P1 neuron activation promotes a latent, internal state that facilitates aggression and courtship, and controls the overt expression of these social behaviors in a threshold-dependent, inverse manner.

*For correspondence: wuwei@caltech.edu

**Competing interests:** The authors declare that no competing interests exist.

## Introduction

Aggression is an innate social behavior that is critical for survival and reproduction in most sexually propagating metazoan species (*Lorenz, 1966*). It serves to establish dominance, or to defend or obtain resources, territory or mating partners. While it is part of most animals' normal behavioral repertoire, maladaptive aggression in humans takes an enormous toll on society (*Filley et al., 2001*; *Miczek et al., 2007*). Yet, we know remarkably little about how the evolutionarily ancient capacity to fight is hardwired into the brain. How are aggression circuits functionally organized? Does the apparent conservation of this behavior across phylogeny reflect a conservation of the underlying neural circuitry? And how is the organization of this circuitry related to that of mating, a closely related social behavior (*Newman, 1999*; *Veening et al., 2005*; *Yang and Shah, 2014*)?

Classical experiments in cats and rodents revealed the existence of localized 'centers' in the medial hypothalamus, whose artificial electrical stimulation was sufficient to trigger attack behavior (*Hess, 1928*; *Hess and Brügger, 1943*; reviewed in *Siegel et al., 1999*; *Kruk, 2014*). Recently, the neurons responsible for this activity were identified in the murine ventromedial hypothalamus (VMH), using steroid hormone receptors as molecular markers, and optogenetics (*Lin et al., 2011*; *Lee et al., 2014*) or genetically targeted cell ablation (*Yang et al., 2013*; reviewed in *Falkner and Lin, 2014*; *Kennedy et al., 2014*). Interestingly, although these cells number ~2,000 per hemisphere, they play a role in male-female mating behavior as well as in fighting (*Lin et al., 2011*; *Sano et al., 2013*; *Yang et al., 2013*; *Lee et al., 2014*). Resolving the functional relationship between neurons in

**eLife digest** For most animals, mating and fighting are critical for survival and reproduction. These behaviors are also closely related and share similar actions. How are such complex behaviors hard-wired into the brain? A fruit fly called *Drosophila melanogaster* is an excellent system to investigate this problem, because flies mate and fight, and powerful genetic tools are available to probe the circuits of neurons that control these behaviors.

A great deal has been learned recently about the neural circuits that control mating, but much less was known about how the circuits for aggression are organized. Hoopfer et al. systematically activated different sets of neurons in thousands of male flies to try to find the circuits that trigger aggression. While this identified some neurons that specifically promoted aggression, it also uncovered a cluster – called P1 neurons – that promoted both aggression and courtship. This was unexpected, because P1 neurons were previously thought to only control courtship behavior.

The P1 neurons produced different behaviors at different stimulation thresholds, with the neurons requiring a stronger level of activation to promote courtship instead of aggression. Moreover, the P1 neurons triggered a lasting change in the internal state of the male that increased his tendency to engage in aggression or courtship. These results are reminiscent of observations made in mice, suggesting small groups of neurons that control mating and fighting may represent an evolutionarily conserved neural circuit "motif" for the control of social behavior.

The next step is to figure out how P1 neurons trigger a persistent internal state of arousal or motivation, and to determine whether and how this circuitry participates in the "decision" to engage in mating or fighting.

this population has been challenging, however, because of their numerosity, and complex connectivity with other brain regions (reviewed in *Swanson, 2000*; *Swanson, 2005*; *De Boer et al., 2015*; *Miczek et al., 2015*).

*Drosophila melanogaster* presents an attractive alternative model for dissecting the neural circuitry of aggression, because of the reduced complexity of its nervous system and the availability of genetic tools for marking and manipulating specific neuronal cell types (*Baier et al., 2002*; *Kravitz and Huber, 2003*; *Zwarts et al., 2012*). *Drosophila* males exhibit robust aggressive behavior, which consists of different agonistic actions such as wing threat, lunging, tussling and boxing (*Dow and von Schilcher, 1975*; *Skrzipek et al., 1979*; *Hoffmann, 1987*; *Chen et al., 2002*). Aggression in flies, as in other species, reflects competition over resources such as food, territory or females (*Hoffmann and Cacoyianni, 1990*; *Chen et al., 2002*; *Hoyer et al., 2008*; *Lim et al., 2014*; *Yuan et al., 2014*), and is under genetic control (*Dierick and Greenspan, 2006*; *Zwarts et al., 2011*).

Relatively little is known about the neurons that specifically control aggression in *Drosophila*. Small groups of cells that release neuromodulators such as octopamine (*Zhou et al., 2008*; *Certel et al., 2010*), dopamine (*Alekseyenko et al., 2013*), serotonin (*Alekseyenko et al., 2014*) or neuropeptide F (*Dierick and Greenspan, 2007*) to regulate levels of aggressiveness have been identified, as have pheromone receptor-expressing neurons controlling this behavior (*Wang and Anderson, 2010*; *Wang et al., 2011*; *Andrews et al., 2014*; reviewed in *Fernández and Kravitz, 2013*). The male-specific form of the sex-determination gene *fruitless (fru$^M$)* is required for normal intermale aggression in *Drosophila* (*Vrontou et al., 2006*) and genetic feminization of relatively broad neuronal populations can feminize (*Nilsen et al., 2004*) aggressive behavior (*Certel et al., 2007*; *Chan and Kravitz, 2007*; *Mundiyanapurath et al., 2009*). However, in comparison to male courtship, where rapid progress has been made in identifying multiple Fru$^M$-expressing cell types that control this behavior (reviewed in *Dickson, 2008*; *Yamamoto and Koganezawa, 2013*; *Yamamoto et al., 2014*), aggression circuitry is more poorly understood. Recently, through a small-scale screen of neuropeptide-expressing neurons (*Hergarden et al., 2012*; *Tayler et al., 2012*), we identified a population of ~3–5 sexually dimorphic Fru$^M$ neurons that control male aggression, via release of *Drosophila* tachykinin (DTK) (*Asahina et al., 2014*). Importantly, manipulation of that cell

population had no influence on courtship behavior. Thus, DTK$^{FruM}$ neurons represent the first subclass of Fru$^M$-expressing neurons that appears specific to the control of aggression.

*Drosophila* affords the ability to perform large-scale screens to identify neurons that control a behavior of interest (e.g., *Rodan et al., 2002*; *von Philipsborn et al., 2011*; *Branson, 2012*; *Seeds et al., 2014*; reviewed in *Simpson, 2009*). However, this approach requires high-throughput behavioral assays (*Wolf et al., 2002*; *Branson et al., 2009*), and aggression is typically scored manually (*Chen et al., 2002*). Recently, however, automated computer algorithms for quantifying aggression in pairs of male flies have been developed (*Hoyer et al., 2008*; *Dankert et al., 2009*; *Kabra et al., 2013*; *Asahina et al., 2014*), overcoming this obstacle.

Here, we describe the first large-scale neuronal activation screen for aggression neurons in *Drosophila*. Using the thermosensitive ion channel dTrpA1 (*Hamada et al., 2008*), we screened a collection of over 3,000 GAL4 lines (*Pfeiffer et al., 2008*; *Jenett et al., 2012*) for flies that exhibited increased fighting following thermogenetic neuronal activation. Among ~20 hits obtained, three exhibited both increased aggression and male-male courtship behavior. Intersectional refinement of expression patterns using split-GAL4 (*Luan et al., 2006*; *Pfeiffer et al., 2010*) indicated that both social behaviors are controlled, in all three hits, by a subpopulation of ~8–10 P1 neurons per hemibrain (*Lee et al., 2000*; *Kimura et al., 2008*). P1 cells are male-specific, Fru$^M$ interneurons that integrate pheromonal (*Kohatsu et al., 2011*; *Clowney et al., 2015*; *Kallman et al., 2015*) and visual (*Pan et al., 2012*; *Kohatsu and Yamamoto, 2015*) cues to promote male courtship behavior (*von Philipsborn et al., 2011*; *Inagaki et al., 2013*; *Bath et al., 2014*). Our results indicate, surprisingly, that at least a subset of P1 neurons, previously thought to control exclusively courtship, can promote male aggression as well. Moreover, we show that they exert this influence by inducing a persistent fly-intrinsic state, lasting for minutes, that enhances these behaviors. These data define a sexually dimorphic neural circuit node that may link internal states to the control of mating and fighting, and identify a potentially conserved circuit 'motif' for the control of social behaviors.

## Results

### A large-scale neuronal activation screen yields multiple independent hits promoting both male wing-extension and aggressive behavior

To find neurons that promote aggressive behaviors we conducted an anatomically unbiased thermogenetic behavioral screen for increased aggression between non-aggressive group-housed (*Wang et al., 2008*) males. We screened a large collection of GAL4 lines with molecularly defined enhancers (*Pfeiffer et al., 2008*; *Jenett et al., 2012*) that directed expression of the temperature-sensitive cation channel *Drosophila* TrpA1 (dTrpA1) (*Hamada et al., 2008*) in different populations of neurons. Multiple pairs of males for each genotype were screened in a high-throughput aggression assay (*Asahina et al., 2014*) using CADABRA software (*Dankert et al., 2009*) for automated scoring of aggressive behaviors (lunging, tussling and wing threat) as well as unilateral wing extension (henceforth referred to simply as wing extension), a component of male courtship song. Of the 3038 lines screened (*Figure 1A*), 815 lines had phenotypes that interfered with the ability to engage in social behaviors. Of the remaining lines, 19 showed a significant increase in one or more aggressive behaviors, compared to an enhancer-less GAL4 control line BDPG4U (pBDPGAL4u; *Pfeiffer et al., 2010*) (*Figure 1A and B*), which does not drive expression in the central nervous system (CNS) (*Figure 1—figure supplement 1D*).

Interestingly, three of these aggression-promoting lines, including R15A01-GAL4, R71G01-GAL4 and R22G11-GAL4, also displayed a robust increase in courtship behavior as measured by wing extension (*Figure 1B*, inset; *Video 1*). These lines drew our attention because of the close relationship between mating and aggression circuits in vertebrates (*Lin et al., 2011*; *Yang et al., 2013*; *Hong et al., 2014*; *Lee et al., 2014*; reviewed in *Newman, 1999*; *Anderson, 2012*; *Kennedy et al., 2014*; *Yang and Shah, 2014*), as well as previous data linking reciprocal control of these behaviors to biogenic amines in *Drosophila* (*Certel et al., 2007*; *Certel et al., 2010*; *Andrews et al., 2014*).

Re-testing of these hits confirmed the dual phenotypes observed in the primary screen (*Figures 1D* and *Figure 1—figure supplement 2D*). Examination of the GAL4 expression patterns in the male nervous system of each line using a UAS-myr::GFP reporter (pJFRC12-10XUAS-IVS-myr:: GFP; *Pfeiffer et al., 2010*) showed that all three lines labeled multiple classes of neurons

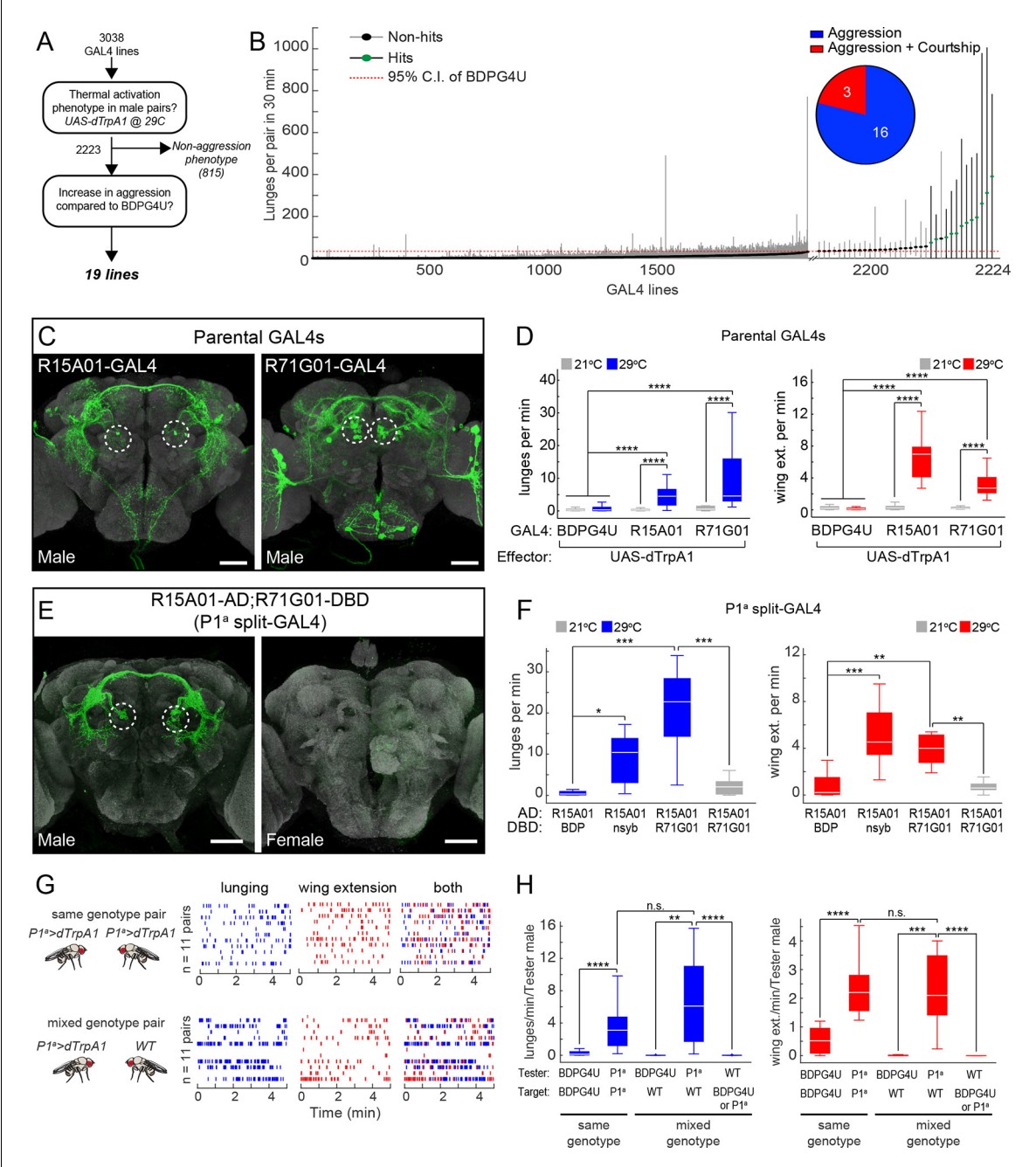

**Figure 1.** Thermogenetic activation of P1 neurons promotes inter-male aggression and wing extension. (**A**) Overview of thermogenetic activation screen to identify GAL4 lines that promote aggression. (**B**) Lunge frequency across the 2223 GAL4 lines that were analyzed for increased aggression. The GAL4 lines are sorted by median lunge frequency. Dots indicate median lunges per pair in 30 min, and lines represent 1.5 times the interquartile range (IQR). For comparison, the dashed red line shows the upper bound of the 95% bootstrap confidence interval (CI) of the BDPG4U>dTrpA1 control males. The blowup on the right side shows the 35 lines that had a median lunge frequency greater than upper bound of the BDPG4U C.I. Green dots indicate lines that showed a significant elevation in lunging compared to BDPG4U after manual verification of lunge scores; black dots were not significantly different. (*Inset*) The proportion of hits that show aggression exclusively (blue) and aggression plus unilateral wing extension (red). Statistical tests are described in Materials and methods. n=226 pairs for BDPG4U>dTrpA1 controls; n=12–24 for each GAL4 line. (**C**) Anatomy of the parental GAL4 lines in the male brain. Confocal images of the central brain for R15A01-GAL4 and R71G01-GAL4. Dashed white circles denote P1 cell bodies. The expression patterns in ventral nerve cord and female nervous system are shown in ***Figure 1—figure supplement 1A-B***. (**D**) Frequency of lunges (left) and wing extensions (right) per minute by pairs of parental GAL4 males expressing dTrpA1 at 21°C (gray) or 29°C (blue or red bars). Pairs tested at 21°C and 29°C: BDPG4U, n=82 and 47; R15A01-GAL4, n=58 and 34; R71G01-GAL4, n=20 and 29. (**E**) Anatomy of split-GAL4 (spGAL4) intersection between R15A01-AD and R71G01-DBD (referred to as 'P1ᵃ') in the male (left) and female (right) brain. Expression in the VNC is shown in

*Figure 1 continued on next page*

*Figure 1 continued*

*Figure 1—figure supplement 1C*, and additional intersections that label P1 neurons are shown in *Figure 1—figure supplements 2* and *3* and summarized in *Supplementary file 1*. (F) Frequency of lunges (left) and wing extensions (right) per min by pairs of males of the indicated spGAL4 combinations with dTrpA1 at 29°C (blue or red bars) or 21°C (gray bars). nsyb refers to a promoter fragment from the *n-Synaptobrevin* gene that labels the majority of neurons in the brain (R57C10-GAL4; *Pfeiffer et al., 2008*) and BDP refers to an enhancerless-DBD lines (pBPZpGAL4DBDU; *Pfeiffer et al., 2010*). n=8–12 pairs. (G) Comparison of lunging and wing extension behaviors by pairs of males of the same genotype (top) or 'mixed genotype' pairs (bottom) in which a BDPG4U>dTrpA1 or P1ᵃ>dTrpA1 male was paired with a WT male. All males were group-housed to reduce baseline aggression. Raster plots showing bouts of lunging (left, blue ticks) and wing extension (middle, red ticks) and both (right) at 29°C by P1ᵃ>dTrpA1 males towards another P1ᵃ>dTrpA1 male or a WT male. Raster plot shows minutes 10–15 of a 30 min assay. (H) Rate of lunging (left) and wing extension (right) per min by the 'Tester' male towards the 'Target' male in same and mixed-genotype pairs (G). Data plotted are from the entire 30 min assay. n=11 for same and mixed-genotype assays. Immunostaining in C and E against GFP is shown in green and neuropil staining with anti-Bruchpilot is shown in light gray. Scale bars in panels C and E are 50 µm. Boxplots (D, F and H) and throughout show the median (white line) flanked by the 25th and 75th percentiles (box) and whiskers showing 1.5 times the IQR. Outliers were excluded from plots for clarity but not from statistical analyses (see Materials and methods). Statistical tests are described in the Materials and methods. *P*-values were adjusted for multiple comparisons by Bonferroni correction. Here and throughout, *: *P*<0.05, **: *P*<0.01, ***: *P*<0.001 and ****: *P*<0.0001. Complete genotypes of the flies used in each figure can be found in *Supplemental file 2*.

The following figure supplements are available for figure 1:

**Figure supplement 1.** Anatomical expression of parental GAL4s and P1ᵃ split-GAL4.

**Figure supplement 2.** Additional split-GAL4 intersections that target P1 neurons.

**Figure supplement 3.** dsx+/R71G01+ P1 neurons drive both aggression and wing extension.

(*Figure 1C*, *Figure 1—figure supplements 1A, B* and *2A*, see also *Supplementary file 1*; R15A01-GAL4: 80–100 neurons; R71G01-GAL4: 100–150 neurons; R22G11-GAL4: >500 neurons). Interestingly, in all three lines (but most clearly R15A01-GAL4 and R71G01-GAL4) these included a population of neurons with cell bodies in the posterior medial protocerebrum (*Figure 1C*, dashed circles and *Figure 1—figure supplement 2A*), and with arborizations in both a prominent commissure and the lateral protocerebral complex. Morphologically, these neurons appeared similar to P1 neurons (*Lee et al., 2000*; *Kimura et al., 2008*), also known as pMP4/-e neurons (*Cachero et al., 2010*; *Yu et al., 2010*), which have been shown to promote courtship behavior (*von Philipsborn et al., 2011*; *Pan et al., 2012*; reviewed in *Yamamoto and Koganezawa, 2013*; *Yamamoto et al., 2014*). We confirmed that these neurons are male-specific and that a subset of them expressed the male isoform Fruᴹ (*Figure 1—figure supplement 1A and B*).

## A subset of P1 neurons promotes both aggression and wing extension

To investigate whether lines R15A01-GAL4, R71G01-GAL4 and R22G11-GAL4 labeled the same population of P1 neurons, and if so, whether those neurons were responsible for the aggression phenotype, we used the split-GAL4 (spGAL4) intersection strategy (*Luan et al., 2006*; *Pfeiffer et al., 2010*) in which GAL4 activity is reconstituted in cells co-expressing the activation domain (AD) and DNA-binding domain (DBD) of GAL4. We generated spGAL4 AD and DBD 'hemi-driver' transgenic flies using the enhancers from each of the three parental lines and tested the different combinations for expression by crossing to UAS-myr::GFP. Indeed, the spGAL4 intersection of R15A01-AD

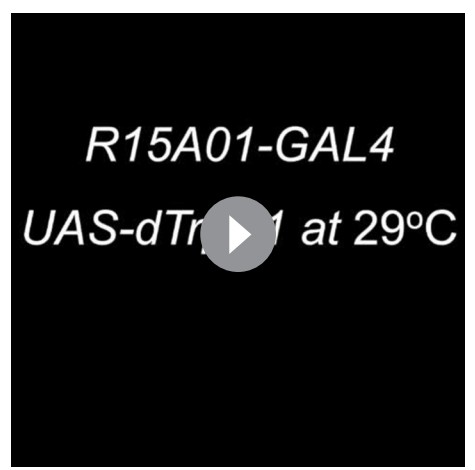

**Video 1.** dTrpA1 activation of R15A01-GAL4. Pairs of R15A01>dTrpA1 males at 29°C exhibit interspersed bouts of aggression (lunging) and courtship (unilateral wing extension).

and R71G01-DBD (henceforth referred to as P1$^a$ spGAL4) labeled approximately 8–10 Fru$^M$-expressing P1 neurons per hemibrain (see also *Inagaki et al., 2013*); no labeling was observed in the ventral nerve cord (VNC) (*Figure 1E* and *Figure 1—figure supplement 1C*). This number of labeled P1 neurons was similar to that in the parental R15A01-GAL4 line (*Supplementary file 1*). In addition to P1 neurons, the P1$^a$ spGAL4 driver labeled a minor and variable population of neurons (<4 per hemibrain) with cell bodies in the lateral horn (LH) that were neither Fru$^M$-positive nor male-specific (*Figure 1—figure supplement 1C*). Other hemi-driver intersectional combinations with R22G11-DBD showed a similar labeling pattern (*Figure 1—figure supplement 2B and C*). Thus, three independent hits from our screen of ~3,000 GAL4 lines labeled a common subset of P1 neurons.

Strikingly, dTrpA1 activation of P1$^a$ spGAL4 males resulted in a robust, temperature-dependent increase in both aggression and wing extension (*Figure 1F*), compared to a control intersection of R15A01-AD and a driverless-DBD line (BDP-DBD). Other spGAL4 combinations (with R22G11-DBD) tested showed a similar increase in both behaviors (*Figure 1—figure supplement 2E and F*). Closer examination of the time course of the interactions between pairs of P1$^a$>dTrpA1 males revealed that the flies exhibited interspersed bouts of aggression and wing extension throughout the duration of the observation period (*Figure 1G*, top; *Video 2*). To verify that the P1 neurons and not the LH population were responsible for the behavioral phenotypes, we used an independent intersectional strategy that targeted P1 but not the LH neurons (*Pan et al., 2012*; see *Figure 1—figure supplement 3A-C*). This intersection also yielded a similar enhancement of aggression and wing extension (*Figure 1—figure supplement 3D*), confirming that activation of P1 interneurons promotes both lunging and wing extension in male pairs.

## Enhanced aggressiveness is a fly-intrinsic response to activation of Fru$^M$+ P1 neurons

The increased lunging caused by thermogenetic activation of P1 neurons could reflect a sustained, fly-intrinsic increase in aggressiveness, or rather a more subtle 'priming' effect amplified by escalating cycles of attack and counter-attack (*Miczek et al., 2007*). To distinguish between these alternatives, we asked whether activation of P1$^a$ neurons could promote unilateral attack towards a non-aggressive target fly. To do this, we paired group-housed P1$^a$>dTrpA1 males with group-housed wild-type (WT) Canton S males, which show little or no aggression (*Wang et al., 2008*). In such mixed-genotype pairs, P1$^a$>dTrpA1 males still showed a significant increase in lunging towards the WT target male throughout the observation period, in comparison to BDPG4U>dTrpA1 controls (*Figures 1G*, lower and 1H). The lunging frequency per P1$^a$>dTrpA1 male was not significantly different between same genotype (P1$^a$ vs P1$^a$) and mixed-genotype pairs (P1$^a$ vs WT) (*Figure 1H*). In contrast, the WT target males (identified by clipping the tip of one wing) showed no significant lunging behavior towards the P1$^a$>dTrpA1 aggressor (*Figure 1H*, left panel, Tester: WT). These data indicate that the sustained increase in aggression caused by activating P1$^a$ neurons in a mixed pair does not depend on counter-attack by the target fly.

It was also possible that the elevated aggression promoted by thermogenetic activation of P1$^a$ neurons was a social response to the concomitantly increased wing extension behavior, which could provide visual and/or auditory cues provoking fighting. However, in mixed genotype pairs, the P1$^a$>dTrpA1 tester flies attacked the group-housed WT target flies despite the fact that the latter exhibited little or no wing extension behavior (*Wang et al., 2008*) (*Figure 1H*, right panel, Tester: WT). In addition, activation of P1$^a$>dTrpA1 males from which both wings had been surgically removed also yielded robust

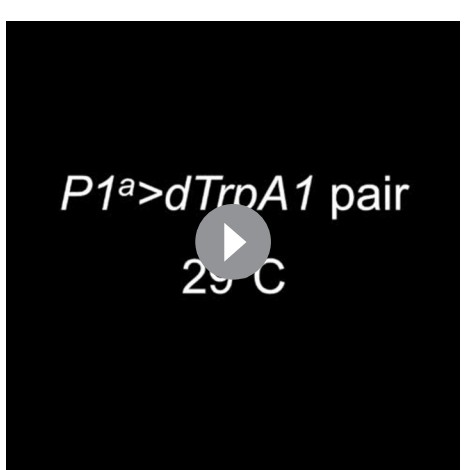

**Video 2.** P1$^a$>dTrpA1 behavioral phenotype. Pairs of P1$^a$>dTrpA1 males at 29°C exhibit interspersed bouts of aggression and wing extension. Note that wing extensions occurring before lunging as shown in this clip is not the invariant sequence of the behaviors (see *Figure 1G*).

fighting behavior (*Video 3*). Finally, activation of other GAL4 lines that promoted increased unilateral wing extension behavior did not cause a concomitant increase in aggression (data not shown). These data indicate that increased male wing extension behavior is neither necessary nor sufficient to explain the increased aggression phenotype. Taken together, these results indicate that the aggression phenotype reflects a fly-intrinsic influence of P1[a] neuron activation.

Not all neurons labeled by the P1[a] spGAL4 line expressed Fru[M] (*Figure 1—figure supplement 1C*, right panel, open triangles). We therefore investigated whether the aggression phenotype was due to activation of Fru[M]–expressing P1 neurons (*von Philipsborn et al., 2011*; *Pan et al., 2012*). To do this, we employed a different intersectional strategy in which dTrpA1 was expressed in the Fru[M]+ subset of neurons labeled by the parental GAL4 drivers R15A01-GAL4 or R71G01-GAL4, using *fru[FLP]* (*Yu et al., 2010*) and UAS-FRT-stop-FRT-dTrpA1[myc] (*Figure 2A*; *von Philipsborn et al., 2011*). This intersection yielded dTrpA1[myc] expression in small and variable number of P1 neurons (1–7 per hemibrain, depending on the parental GAL4 line; *Figures 2B–D*), probably reflecting the relative inefficiency of Fru[FLP]-mediated recombination (*Yu et al., 2010*; *von Philipsborn et al., 2011*).

Despite the small number of P1 neurons labeled, all of these intersectional flies exhibited a significant increase in lunging upon dTrpA1 activation (*Figures 2E and F*). In contrast to the enhanced aggression, however, there was little to no enhancement of wing extension (*Figures 2E and G*), consistent with earlier reports that at least 10 P1 neurons must express dTrpA1[myc] to significantly increase singing relative to controls (*von Philipsborn et al., 2011*). These observations indicate that activation of just a few Fru[M] P1 neurons is sufficient to increase aggression. They also demonstrate that the increased aggression phenotype can be observed in the absence of elevated wing extension, further dissociating these two behavioral phenotypes of P1 neuron activation.

## Weak optogenetic activation of P1 neurons promotes aggression but not wing extension

The foregoing experiments indicated that increased aggression could be uncoupled from wing extension by activating a small subset of Fru[M]+ P1 neurons, suggesting that these two behaviors might require different threshold levels of P1 activation. However, the effectors used in these experiments were different, UAS-dTrpA1 (*Hamada et al., 2008*) vs. UAS-FRT-stop-FRT-dTrpA1[myc] (*von Philipsborn et al., 2011*), making direct comparisons difficult. To investigate more systematically whether elevated aggression and wing extension could be separated by manipulating the level of activity among P1 neurons, we turned to optogenetic effectors, which afford greater control over the dynamic range of neuronal activation than does dTrpA1 (*Inagaki et al., 2013*). To minimize interference with visual cues, which are important for aggression, we used the red-shifted opsin CsChrimson (*Klapoetke et al., 2014*) at 685 nm, a wavelength to which flies are relatively insensitive, and low light-power density (0.02 mW/mm$^2$). In order to compare the effects of photoactivation on lunging vs. wing extension in the same cohort of flies, we used the P1[a] spGAL4 driver, whose activation using dTrpA1 evoked both social behaviors (*Figure 1F*).

Optogenetic stimulation of *P1[a]>CsChrimson* male pairs was performed over a range of photostimulation (PS) frequencies between 1–50 Hz (*Figure 3A*). To confirm that this range of stimulation frequencies resulted in corresponding increases in P1 activity, we performed 2-photon imaging of calcium transients in P1[a]>GCaMP6s neurons that co-expressed a tdTomato-tagged Chrimson, in an open-cuticle preparation in intact flies (*Inagaki et al., 2012*; *Inagaki et al., 2013*; *Figure 3—figure supplement 1*). We observed a linear increase in the magnitude of

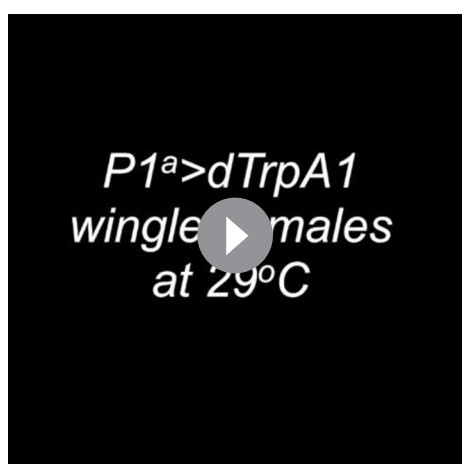

**Video 3.** Activation of P1 neurons induces aggression between wingless males. Pair of P1[a]>dTrpA1 males at 29°C with wings removed still show aggression.

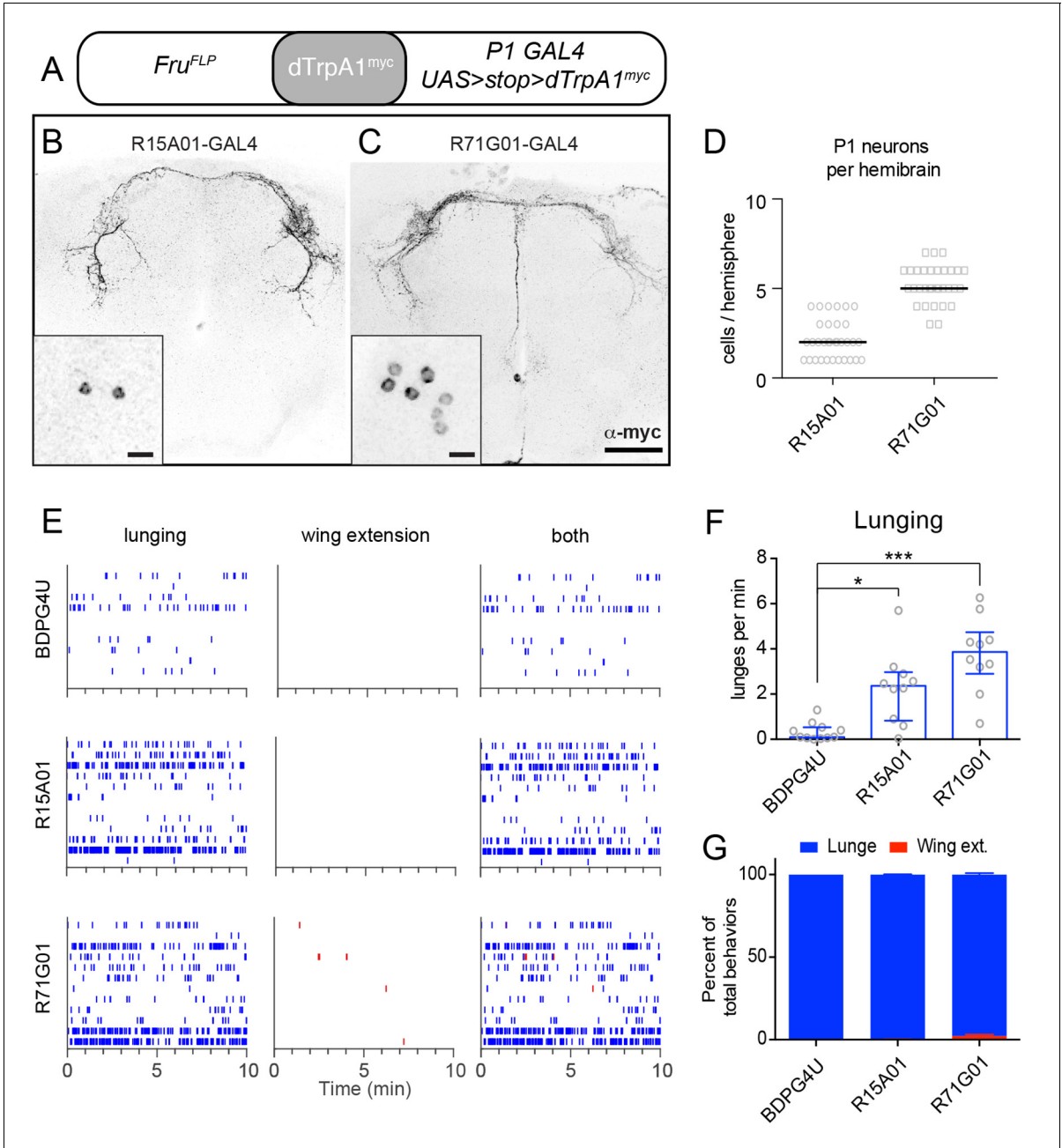

**Figure 2.** TrpA1 activation of Fru^M+ subset of P1 neurons in parental GAL4 lines promotes aggression but not wing extension. (**A**) Intersectional strategy for targeting Fru^M+ neurons in the parental GAL4 lines. (**B, C**), Intersectional expression of dTrpRPA1^myc in male brains of R15A01-GAL4 ∩ Fru^FLP (**B**) and R71G01-GAL4 ∩ Fru^FLP (**C**) flies. Confocal Z-stacks through the anterior brain showing antibody staining against dTrpA1^myc of the P1 neuron projections. Insets show high-magnfication views of the P1 neurons labeled in the intersection in one hemisphere of the brain. Scale bars are 50 μm, and 10 μm in the insets. (**D**) Number of P1 neurons per brain hemisphere labeled with dTrpA1^myc in males of R15A01-GAL4 and R71G01-GAL4 intersections with *fru^FLP*. n=16 brains for each. (**E**) Raster plots of lunging (left), wing extensions (middle) and both (right) in each GAL4 ∩ Fru^FLP intersection during minutes 10-20 of a 30 min aggression assay. (**F**) Number of lunges per min by pairs of males of the indicated genotypes at 32°C. (**G**) Proportion of lunges and wing extensions performed by each fly as a percentage of total behaviors (lunges + wing extensions). Bars are mean ± s.e.m. n=12 pairs per genotype. Asterisks are Bonferroni-corrected *P*-values.

GCaMP6s responses from each P1 neuron as the PS frequency was increased from 10 to 50 Hz.

In freely behaving male fly pairs, as the PS frequency was increased from 1 to 20 Hz, there was a progressive increase in lunging rate by P1^a>CsChrimson vs. BDPG4U>CsChrimson control flies

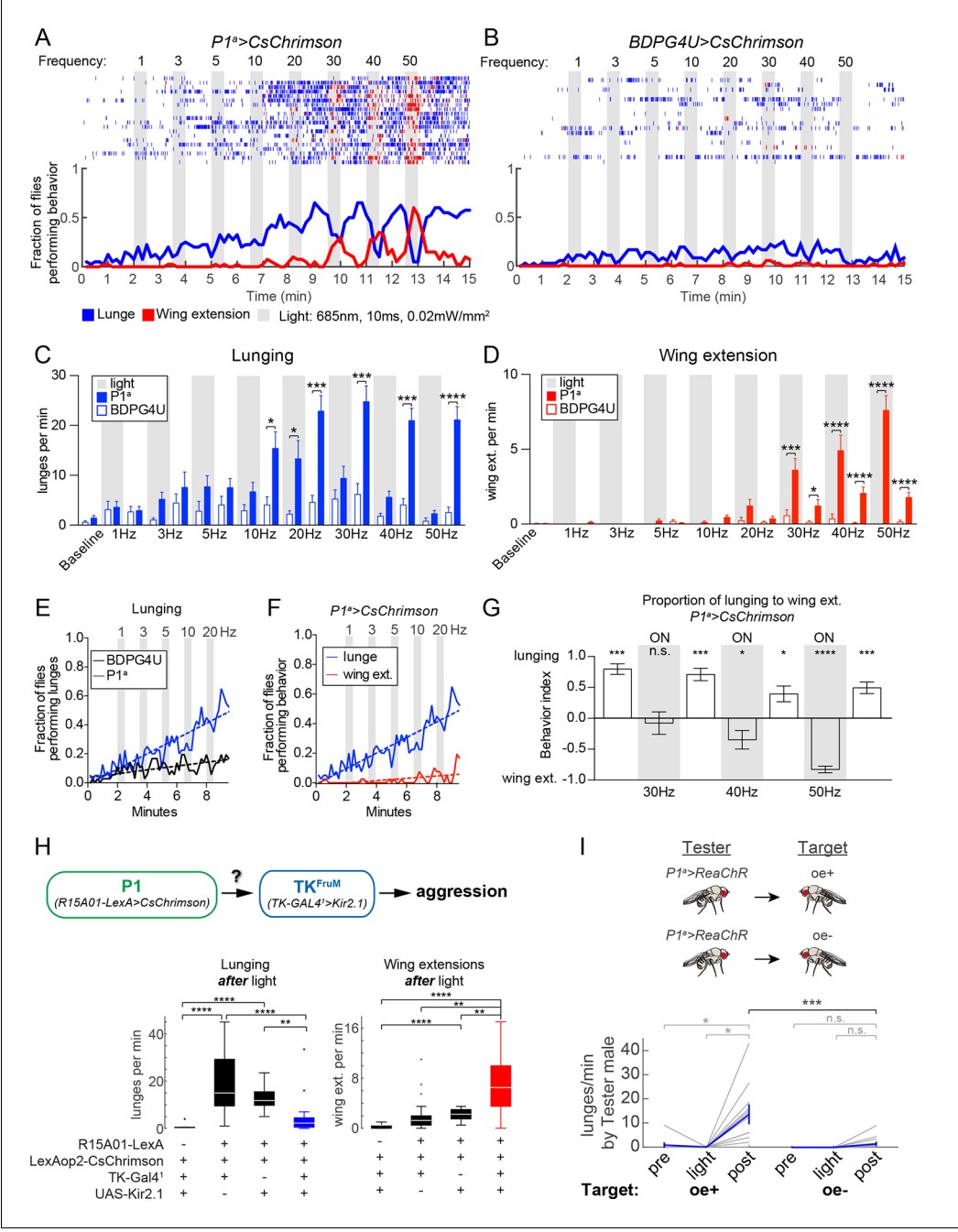

**Figure 3.** Optogenetic stimulation of P1 neurons acutely suppresses aggression and promotes wing extension. (**A–B**) Frequency titration of 685 nm light on pairs of P1$^a$>CsChrimson::venus (**A**) or BDPG4U>CsChrimson::venus (**B**) males. Blocks of 30 s photostimulation (PS, gray bars) with increasing stimulation frequency separated by 1 min intervals were sequentially delivered to the males. *Top*: raster plots showing lunging (blue ticks) and wing extensions (red ticks) by each pair. *Bottom*: fraction of flies performing lunging or wing extension in 10 s time bins. n=20 pairs for P1$^a$>CsChrimson and 18 pairs for BDPG4U>CsChrimson. *Figure 3—figure supplement 1* shows functional imaging of P1 neuronal responses to Chrimson activation over the same range of stimulation frequencies. (**C, D**) Frequency of lunging (**C**) and wing extension (**D**) by P1$^a$>CsChrimson and BDPG4U>CsChrimson male pairs during each stimulation (gray bars) and post-stimulation period. Baseline refers to the 2 min prior to the first PS. Bars are mean ± s.e.m.; asterisks are Dunn's corrected *P*-values for between-genotype comparisons. See *Figure 3—figure supplement 2A* for comparison of P1$^a$>CsChrimson behavior during each interval with pre-stimulation baseline. (**E**) Fraction of flies performing lunges from baseline to the 30 Hz PS period (0 to 9:30 min) in P1$^a$>CsChrimson and BDPG4U>CsChrimson genotypes (data are from the experiments shown in A & B, blue lines). A linear regression was fit to each curve (see Materials and methods). The slope of both the BDPG4U and P1$^a$ linear regressions was significantly different from zero, *P*<0.0001. The fraction of flies performing lunging was calculated as in (**A**). (**F**) Fraction of P1$^a$>CsChrimson males performing lunging (blue; same analysis as in panel (**E**) reproduced for purposes of comparison) or wing extension (red) over the same period. The slopes of the lines are significantly different (*P*<0.0001).
*Figure 3 continued on next page*

*Figure 3 continued*

Data are from the experiment shown in (A), blue and red lines. (G) Proportion of lunges to wing extensions performed by P1$^a$>CsChrimson males from the post-20 Hz through post-50 Hz intervals (8.5–15 min in A). Behavior index = (lunges – wing ext.)/(lunges + wing ext.). Lunging and wing extension rates (bouts per min) for each pair were divided by the maximum lunge and wing extension rates in order to normalize between the two behaviors. Bars are mean ± s.e.m. Asterisks represent significant deviation from zero (Wilcoxon signed-rank test). See also *Figure 3—figure supplement 2C-E* for velocity of pairs and single males. (H) Epistasis experiment between P1 and TK$^{FruM}$ neurons. *Top*: Schematic of experimental genotypes. P1$^a$>CsChrimson was used to activate P1 neurons while silencing TK$^{FruM}$ neurons with TK-GAL4$^1$>UAS::eGFP-Kir2.1. *Bottom*: Lunge (left) and wing extension (right) frequency per pair after PS with continuous 627nm light at 0.2 mW/mm$^2$ for 30 s. n=14–22 pairs per genotype; asterisks are Bonferroni-adjusted *P*-values. See also *Figure 3—figure supplement 3* for raster plots and quantification of behavior during PS. (I) Lunging by photostimulated P1$^a$>ReaChR males ('Tester') towards 'Target' males with intact (oe+) or ablated (oe-) oenocytes. All males were group-housed to reduce baseline aggression. PS was 1 min of 530 nm light (10 Hz, 20 ms pulse-width, 0.2 mW/mm$^2$). Pairs were allowed to interact for 1 min prior to PS ('Pre') and 2 min after PS ('Post'). Gray lines are individual pairs and bold lines are mean ± s.e.m. Asterisks in gray are within genotype comparison and asterisks in black are between genotype comparisons. n=11 pairs (oe+) and 16 pairs (oe-).

The following figure supplements are available for figure 3:

**Figure supplement 1.** Functional imaging of P1 neuronal responses to optogenetic activation.

**Figure supplement 2.** Photoactivation of P1$^a$>CsChrimson flies.

**Figure supplement 3.** Effect of silencing TK$^{FruM}$ neurons on behavioral phenotype of P1 neuron optogenetic activation.

(*Figures 3A and B*). Linear regression fits of the fraction of flies that lunged as a function of time (*Figure 3E*) showed a significant difference between the slopes for P1$^a$ vs. BDPG4U males (*P*<0.0001). At 10 Hz and above (the threshold frequency that yielded a significant increase in GCaMP6 responses; *Figure 3—figure supplement 1*) the lunging rate was significantly higher than that of genetic controls stimulated at the same frequency (*Figure 3C*, blue vs. white bars), and of pre-stimulation baseline lunging levels in the experimental genotype (*Figure 3—figure supplement 2A*). However, the enhancement of aggression in P1$^a$>CsChrimson flies was not time-locked to the onset of PS, but rather tended to increase following PS offset (*Figure 3C* and *Figure 3—figure supplement 2B*).

Importantly, across this same frequency range (1–20 Hz) there was little promotion of wing extension (*Figures 3A and D*, red lines/bars; *Figure 3F*, red lines). As the PS frequency was increased to 30 Hz and higher, however, optogenetic activation evoked robust wing extension during the light ON phase (*Figure 3D*, red bars). The rate of wing extension progressively increased from 30 to 50 Hz (*Figure 3D*, shaded regions, red bars; *Figure 3—figure supplement 2C*). These data indicate that the threshold for optogenetic enhancement of aggression by P1$^a$ neurons is lower than that required for wing extension, under these conditions. Similar results were obtained using a different optogenetic effector, ReaChR (*Inagaki et al., 2013*; *Lin et al., 2013*) (see *Figures 4A and D* below).

## Strong activation of P1 neurons reciprocally controls aggression and wing extension

As the PS frequency was raised above the threshold for evoking wing extension, we observed that aggression was increasingly suppressed during the light ON phase (*Figure 3C*, 30–50 Hz, shaded regions; *Figure 3G*, 'ON'; *Video 4*). Following PS offset, however, aggression rapidly resumed and wing extension was concomitantly suppressed (*Figures 3A and C*, 30–50 Hz, blue bars; *Figure 3G*; see also *Figure 3—figure supplement 2E* for 50 Hz PS; *Video 4*). The inhibition of aggression during PS was unlikely due to an effect of the light itself, as flies are relatively insensitive to the red wavelength used (685 nm) and aggression was not suppressed during PS at frequencies ≤20 Hz (*Figure 3C* and *Figure 3—figure supplement 2B*). Therefore, strong activation of wing extension may physically interfere with or otherwise suppress aggression. We also observed that high-frequency P1 activation promoted extended periods of locomotor arrest, which was relieved following PS offset (*Figure 3—figure supplement 2C-E*). As lunging requires active locomotion, such immobility may physically prevent it. By contrast, wing extension during PS occurred frequently in stationary flies (*Video 4*).

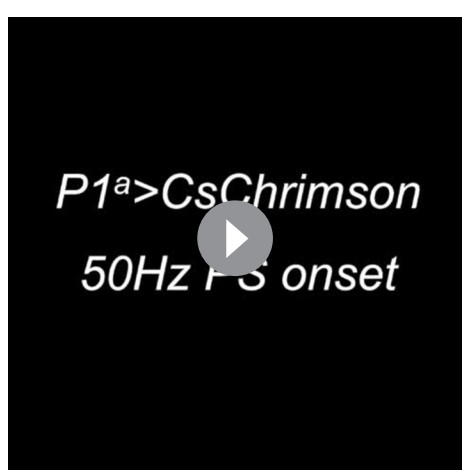

**Video 4.** P1ᵃ>CsChrimson phenotype in pairs of males during strong photostimulation. Pairs of P1ᵃ>CsChrimson males at the onset of 50 Hz PS (see *Figure 3*) show a rapid suppression of aggression and locomotion and an increase in wing extensions. At the offset of the PS, males switch back to aggression.

Conversely, we observed a relatively rapid decline of wing extension following the offset of high-frequency PS (*Figures 3A and D*, red; 3G and *Figure 3—figure supplement 2E*; $t_{1/2} = 9 \pm 3$ s), concomitant with the re-initiation of aggression. This decline contrasted with the persistent wing extension observed following transient P1 activation in solitary flies (*Inagaki et al., 2013*; *Bath et al., 2014*) (*Figure 3—figure supplement 2E*, single males, $t_{1/2} = 56 \pm 7$ s). Therefore, we wondered whether it was caused by aggression per se, or rather reflected some independent parallel influence to suppress wing extension following stimulus offset. To address this possibility, we performed an epistasis experiment (*Figure 3H*) in which optogenetic P1 activation was performed while simultaneously inhibiting a population of *Drosophila* tachykinin-expressing, Fru[M] neurons (TK[FruM] neurons; a subset of so-called aSP6 Fru[M] neurons; *Yu et al., 2010*; *Kohl et al., 2013*) previously shown to be necessary and sufficient for aggression (*Asahina et al., 2014*).

Indeed, silencing of TK[FruM] neurons using Kir2.1 (*Baines et al., 2001*) strongly suppressed the enhancement of lunging that occurred following termination of P1 photoactivation (*Figure 3H*, lunging, blue box and *Figure 3—figure supplement 3A*, bottom raster). This result indicates that TK[FruM] neurons are either functionally downstream of P1 neurons, or act in a parallel pathway required for aggression. Strikingly, although silencing TK[FruM] neurons had no effect to diminish or enhance wing extension during PS (*Figure 3—figure supplement 3A and B*), it caused a strong and statistically significant increase in wing extension during the light OFF period (*Figure 3H*, right, red box; *Figure 3—figure supplement 3A*, bottom raster). In other words, silencing of TK[FruM] neurons in male pairs allowed P1ᵃ activation-induced wing extension to persist into the post-stimulation period, similar to the response of solitary flies (*Inagaki et al., 2013*). These data indicate that the rapid decrease in wing extension observed following the offset of photostimulation (*Figure 3A* and *Figure 3—figure supplement 2E*) is indeed caused by the resumption of aggression.

To investigate further why the effect of P1 activation switches from promoting wing extension to aggression following photostimulation offset, we asked whether this switch was dependent on male-specific chemosensory cues (*Fernandez et al., 2010*; *Wang and Anderson, 2010*; *Wang et al., 2011*). To investigate this possibility, P1ᵃ>ReaChR males were paired with target males lacking oenocytes (oe-), the cells which produce cuticular hydrocarbon pheromones (*Billeter et al., 2009*). Under these conditions, a significant reduction in the frequency of lunging following transient P1ᵃ photoactivation was observed, in comparison to P1ᵃ>ReaChR males paired with control male targets bearing intact oenocytes (oe+) (*Figure 3I*). Thus, the abrupt switch from enhanced wing extension to enhanced fighting following the offset of P1ᵃ activation requires appropriate chemosensory cues on the conspecific target.

## P1ᵃ neuron activation triggers a persistent internal state that enhances aggressiveness

The fact that ongoing aggression was observed in the 1 min intervals between successive episodes of high-frequency P1ᵃ neuron activation (*Figure 3A*) indicated that this behavior, once triggered, could be sustained at least briefly without continued PS. To investigate whether this persistent fighting required repeated cycles of photostimulation, or rather could occur following a single high-intensity PS trial, we exposed P1ᵃ>ReaChR male pairs to one 1-min episode of photoactivation. Under these conditions, wing extension was robustly induced during PS, but switched to aggression following PS offset; this lunging persisted for at least 10 min (*Figures 4A—C*). Such robust, persistent

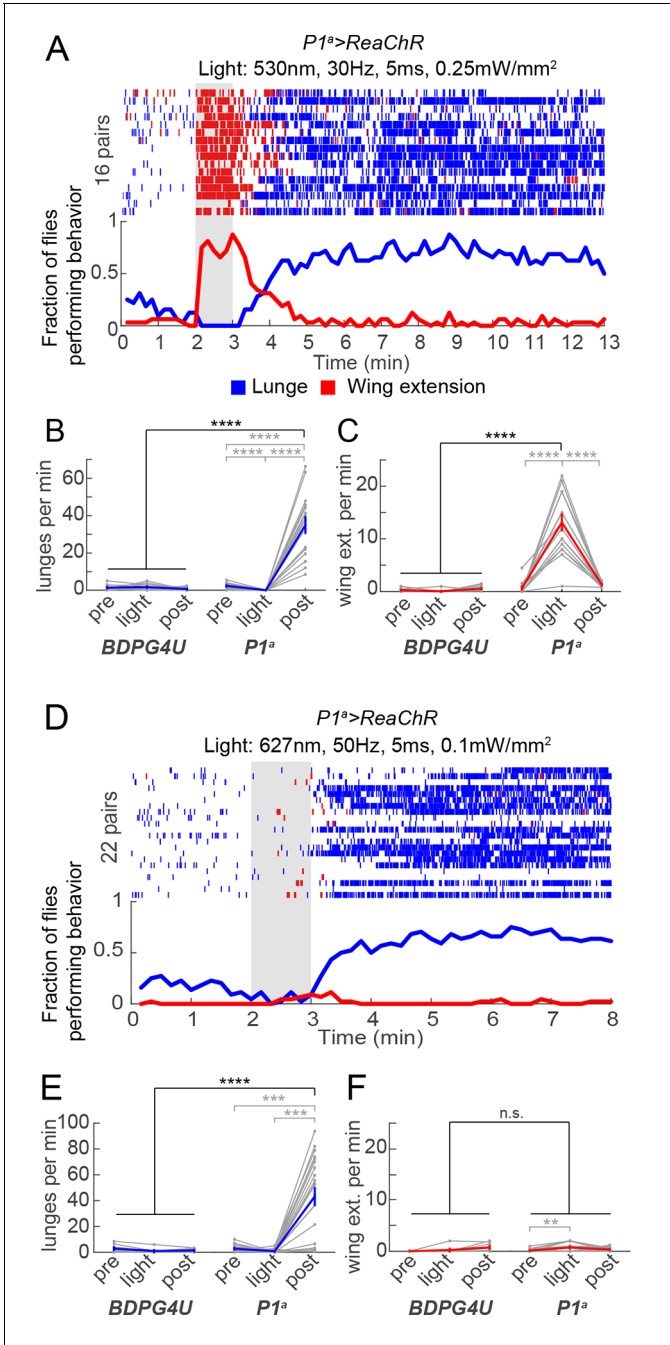

**Figure 4.** Transient activation of P1 neurons induces persistent aggression. (**A**) Activation of P1ᵃ spGAL4 with UAS-ReaChR. Pairs of males were photostimulated for 1 min with 530 nm light (30 Hz, 5 ms pulse-width, 0.25 mW/mm²). Plot properties are the same as in *Figure 3A*. (**B, C**) Lunges (**B**) and wing extensions (**C**) per minute performed by P1ᵃ>ReaChR and BDPG4U>ReaChR males in the period before ('pre'), during ('light') and after ('post') photostimulation (shown in **A**). Gray lines are individual pairs and bold lines are mean ± s.e.m. Asterisks in gray are within genotype comparisons and asterisks in black are between genotype comparisons. n=15–16 pairs per genotype. (**D**) P1ᵃ>ReaChR males stimulated more weakly than in (**A**) (627 nm, 50 Hz, 5 ms pulse-width, 0.1 mW/mm²) for 60 s. (**E, F**) Lunges (**E**) and wing extensions per minute (**F**) performed by P1ᵃ>ReaChR and BDPG4U>ReaChR males in the PS conditions shown in (**D**). See also *Figure 4—figure supplement 1* for lunging in the cohort of males that exhibited no wing extensions prior to the onset of fighting. Plot properties and statistics are the same as in B and C. n=8 pairs for BDPG4U and n= 22 for P1ᵃ.

The following figure supplement is available for figure 4:

**Figure supplement 1.** Wing extension is not necessary for persistent post-PS aggression.

aggression could also be triggered by weaker photosimulation conditions that promoted only occasional wing extension bouts in a subset of flies (*Figures 4D−F*). Under these latter conditions, persistent post-PS aggression was triggered even in the cohort of fly pairs (8/22) that showed no wing extension at all during the PS period (*Figure 4—figure supplement 1*). Thus, persistent aggression could be promoted by a single episode of P1[a] neuronal activation, and did not depend on male-male wing extension behavior during PS.

To determine whether the persistent aggression initiated by transient photoactivation of P1[a] neurons reflected an enduring fly-intrinsic property, or rather a triggering event perpetuated by social feedback—i.e., iterative cycles of reciprocal attack—we paired P1[a]>ReaChR flies with a group-housed, non-aggressive WT target fly (cf. *Figure 1G*). Under these conditions, persistent attack towards the WT target lasting at least 10

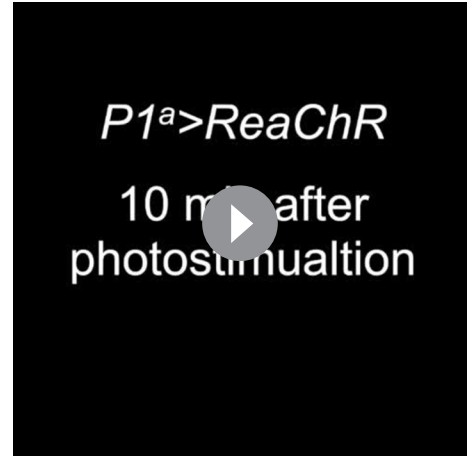

**Video 5.** P1[a]>ReaChR males interacting following barrier removal 10 min after photostimulation. Males were stimulated as described in *Figure 5A*.

min was triggered by a single episode of photostimulation (*Figure 5—figure supplement 1A,B*). These results suggested that persistent aggression was not simply perpetuated by self-sustaining cycles of attack and counter-attack, but likely reflected a long-lived internal state in the tester fly. However, it remained formally possible that the act of attacking and defeating the target fly was behaviorally self-reinforcing, as demonstrated in rodents (*Fish et al., 2002*). We therefore investigated whether such a persistent state required any social interactions at all, or could endure in a latent form in solitary flies.

To address this question, we modified our behavior arena so that the two flies were physically separated from each other during the PS period. We did this by adding a removable opaque barrier across the middle of the arena, thereby isolating the two flies (*Figure 5A*, left; see Materials and methods). We optogenetically activated P1 neurons in such separated males for 1 min, and waited various lengths of time before removing the barrier and allowing the males to interact (*Figure 5A*, right). As expected, separated P1[a]>ReaChR males (*Figure 5B*), but not BDPG4U>ReaChR males (*Figure 5C*), showed a robust increase in wing extension during PS, which persisted after light offset (*Inagaki et al., 2013*). Upon removal of the barrier 1 min after PS offset (a time when the flies were still performing wing extensions), these P1[a]>ReaChR males switched to aggression (*Figures 5B*, blue rasters). Strikingly, enhanced aggression was observed even when removal of the barrier was delayed for 10 min after the offset of PS (*Figures 5D and E*) (*Video 5*), by which time persistent wing extension behavior (*Figure 5F*), as well as locomotor activity (see *Figure 5—figure supplement 1C*), had decayed to baseline levels for at least 5 min. The level of lunging after barrier removal for both the 1 min and 10 min conditions was significantly elevated compared to BDPG4U>ReaChR controls (*Figure 5G*), as well as control P1[a]>ReaChR males that received no PS (*Figure 5H*). Although the latency to lunging was not significantly different between barrier removal at 1 min vs. 10 min (*Figure 5—figure supplement 1D*), the flies took longer to achieve maximal levels of aggression in the latter case, suggestive of a decay in the internal state promoted by P1[a] activation (*Figure 5—figure supplement 1E*). Together, these experiments suggest that P1 neuron activation triggers a persistent internal state facilitating future social behaviors, which can endure for minutes in behaviorally quiescent solitary flies.

## Discussion

Aggression is an innate, complex social behavior whose neural circuit basis is poorly understood. Here, we report the results of the first large-scale systematic screen, in any species, for neurons that promote inter-male aggression when artificially activated. Thermogenetic activation of ~3,000 *Drosophila* GAL4 lines (*Jenett et al., 2012*) using the temperature-sensitive cation channel dTrpA1

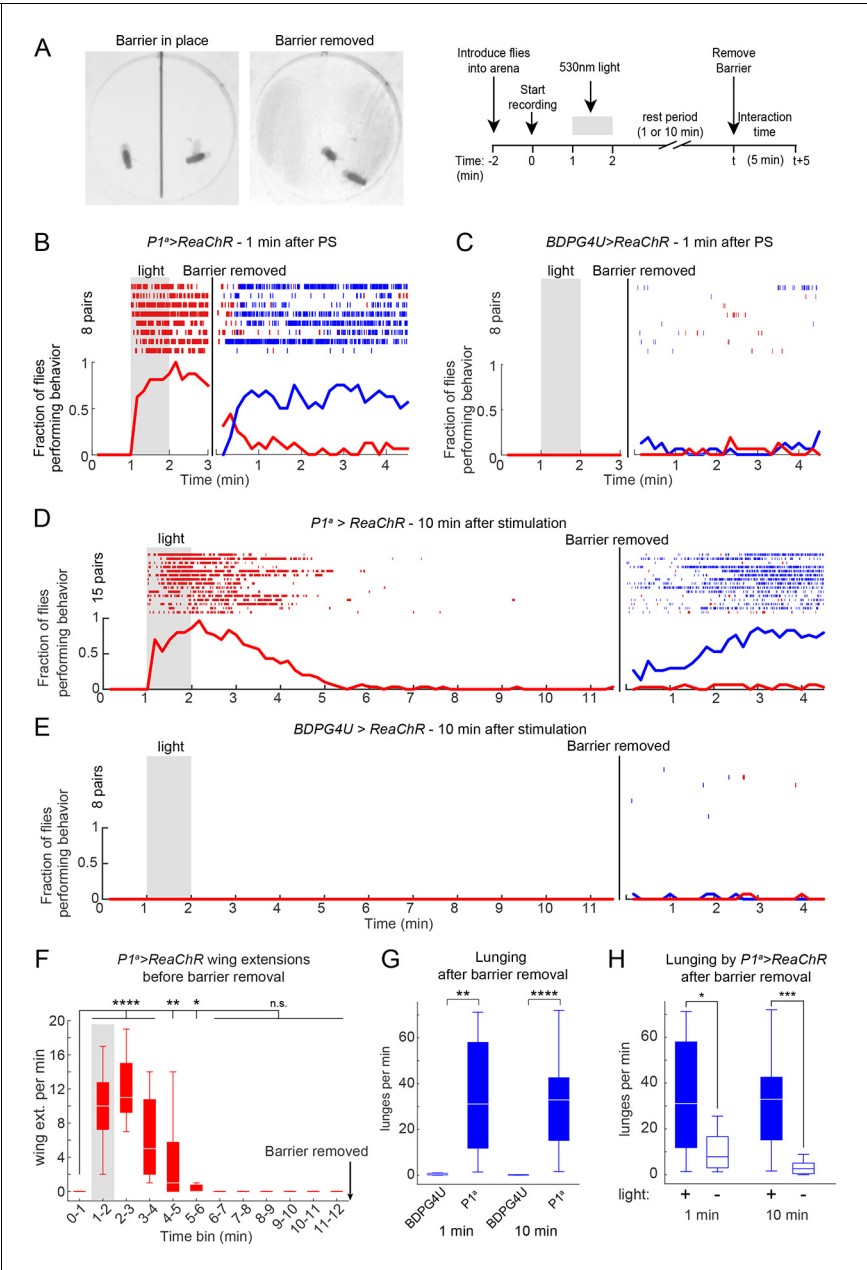

**Figure 5.** P1 activation induces a persistent state that is fly intrinsic. (**A, B**) Assay and experimental design used to separate optogenetic activation from social interaction. Males were placed on opposite sides of a removable opaque barrier (*left*; experimental design on *right*). After acclimation to the arena, a 1 min PS with 530 nm light (10 Hz, 20 ms pulse-width, 0.2 mW/mm²) was delivered to the separated males. The barrier was removed 1 or 10 min after the end of PS and male interactions were recorded for 5 min. (**B–C**) Behavior of P1ᵃ>ReaChR (**B**) and BDPGU>ReaChR (**C**) males when the barrier was removed 1 min after PS (barrier removal is represented by the black line). (**D, E**) Behavior of P1ᵃ>ReaChR (**D**) and BDPG4U>ReaChR (**E**) males when the barrier was removed 10 min after the end of PS. Velocity of P1ᵃ>ReaChR males before the barrier removal is shown in *Figure 5—figure supplement 1C*. (**F**) Wing extensions per minute performed by separated P1ᵃ>ReaChR males in the 10 min time period before barrier removal in the experiment shown in (**D**). (**G**) Lunges per minute performed by pairs of the indicated genotypes after the barrier was removed. Latency to increased aggression is shown in *Figure 5—figure supplement 1D–E*. (**H**) Comparison of lunges per minute after barrier removal for P1ᵃ>ReaChR pairs with or without photostimulation. Asterisks are Bonferroni-adjusted *P*-values.

The following figure supplement is available for figure 5:

**Figure supplement 1.** Persistent aggression after optogenetic activation of P1 neurons.

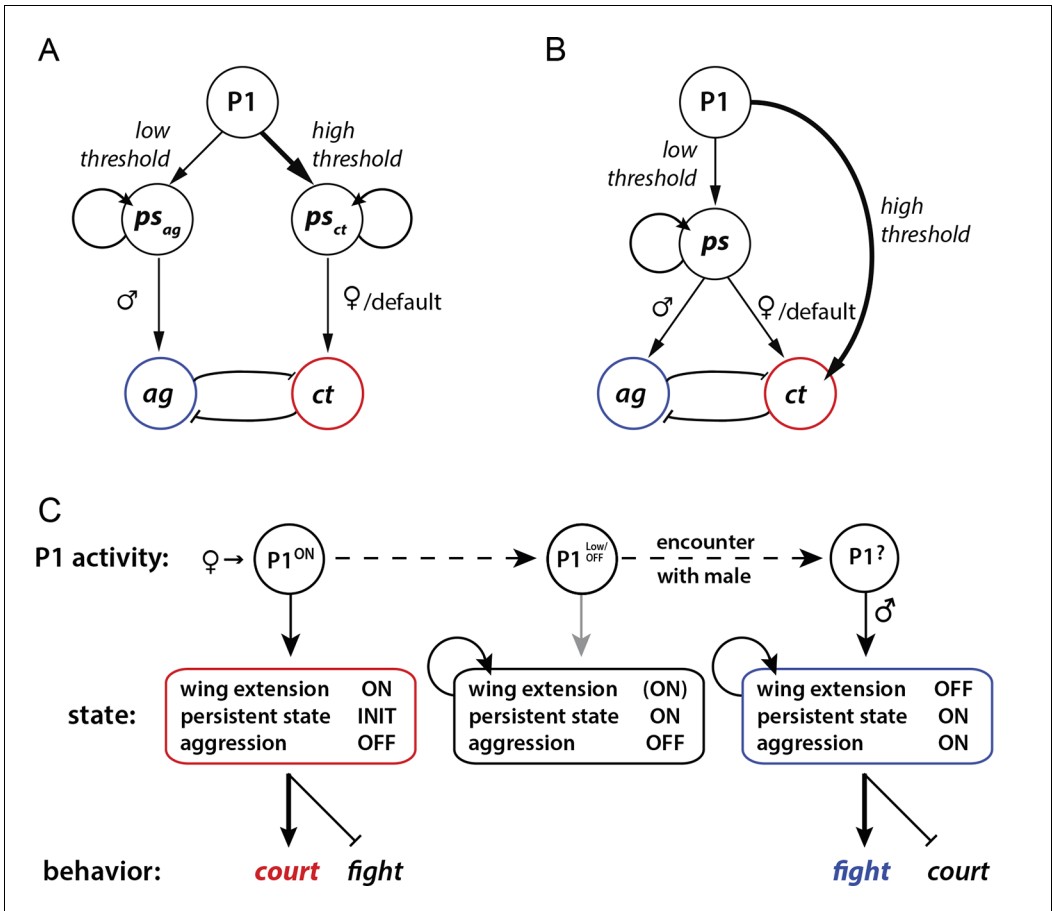

**Figure 6.** Models for how P1 neurons may regulate courtship and aggression. (**A, B**) Alternative models for how P1 neurons influence aggression ('ag') vs. courtship ('ct')-promoting circuitry. (**A**) 'Direct' model. P1 neurons exert parallel and independent influences on downstream circuits that promote persistent internal states specific to aggression ('$ps_{ag}$') or courtship ('$ps_{ct}$'), at low and high thresholds, respectively. These influences may be exerted by common or distinct subpopulations of P1 neurons (not illustrated). Following the offset of phasic P1 activation, the persistent aggression state drives overt fighting behavior in the presence of a male conspecific. In the absence of a male (or presence of a female), persistent courtship behavior is expressed. Reciprocal inhibition between aggression and courtship circuitry is posited to occur at some point downstream of P1 neurons (blunt arrows), although if the P1 population is heterogeneous it could occur within that population as well (not shown). (**B**) 'Indirect' model. Activation of P1 neurons at a relatively low level triggers a persistent internal state ('$ps$') that is neutral with respect to aggression vs. courtship. This state can facilitate either social behavior, depending upon the sex of the conspecific encountered. A parallel, high-threshold pathway that directly activates command modules for courtship song (**von Philipsborn et al., 2011**) is included to account for the effect of acute optogenetic activation of P1 neurons to trigger stimulus-locked wing extension behavior (**Inagaki et al., 2013**; **Bath et al., 2014**). (**C**) State diagram illustrating how male social behavior may be regulated by P1 neuron activity. P1<sup>ON</sup> denotes activation of P1 neurons by a female. Following disengagement from courtship, the persistent internal state triggered by P1 activation is maintained (middle box). A subsequent encounter with a male leads to aggression (right box). P1 neurons may be activated by some male-specific cues, but more weakly than by female cues (**Kohatsu et al., 2011**).

(**Hamada et al., 2008**) yielded ~20 hits exhibiting elevated aggression, indicating a high level of specificity. While the majority of these hits displayed an aggression-specific phenotype (lunging, wing threat, tussling), three of them exhibited a mixture of increased aggression and increased inter-male courtship (measured as unilateral wing extension). Although each of these three hits expressed GAL4 under the control of a different DNA cis-regulatory element (**Pfeiffer et al., 2008**), and labeled different groups of neurons numbering in the hundreds, the expression pattern of all three lines included P1 interneurons. Remarkably, intersectional experiments using the split-GAL4 technique revealed that the P1 neurons were virtually the only cells in common between these three hits, and, moreover, that these neurons were responsible for both the aggression and wing extension phenotypes. The fact that P1 neurons were independently recovered in three independent hits, constituting only 0.1% of the lines initially screened, and could be isolated from 'contaminating' cell

types by carrying out pair-wise intersectional combinations of these hits (*Luan et al., 2006*; *Pfeiffer et al., 2010*), testifies to the reproducibility of the phenotype and the power of the screen to reveal rare populations of neurons that control social behavior in *Drosophila*.

## P1 neurons can promote aggression as well as wing extension

The identification of P1 neurons in a screen for aggression-promoting neurons was surprising, as the prevailing view is that these cells specifically promote male courtship behavior, in response to female-specific pheromones (*Kohatsu et al., 2011*; *von Philipsborn et al., 2011*; *Pan et al., 2012*; *Clowney et al., 2015*; *Kallman et al., 2015*). Indeed, we were initially concerned that the elevated aggression phenotype might simply be an indirect, social response to elevated inter-male wing extensions promoted by P1 activation. However, in mixed-genotype pairs, activation of P1[a] neurons produced aggression towards group-housed wild-type flies that did not exhibit any wing extension. Furthermore, activation with dTrpA1 of a small subset of Fru[M]+ P1 neurons (~2–5 of 20 per hemi-brain) yielded flies that exhibited increased aggression without a concomitant increase in wing extension behavior. These data suggest that activation of P1 neurons promotes aggression in a fly-intrinsic manner, and is not an indirect social response provoked by elevated wing extension behavior.

Our evidence that P1 neurons play a role in aggression is, however, based on gain-of-function experiments. Unfortunately, attempts to demonstrate a requirement for P1 neurons during spontaneous aggression using various methods of cell inactivation or ablation have so far been unsuccessful. These negative results may reflect incomplete penetrance of effector function (*Thum et al., 2006*), expression of the effectors in only a subset of P1 neurons, redundant aggression circuits that act in parallel with P1, or they may indicate that P1 neurons promote aggression under some but not all conditions. Technical limitations in *Drosophila* currently preclude the ability to record neuronal activity from freely behaving flies engaged in social behavior, unlike the case in mice (*Lin et al., 2011*), leaving open the question of whether P1 neurons are normally activated during aggressive encounters. Thus, while artificial activation of P1 neurons can promote aggression, it remains unclear whether they are normally required for this behavior, and if so under what circumstances.

## P1 neurons promote aggression and wing extension in a threshold-dependent, inverse manner

We found that the effect of P1 neurons to promote aggression vs. wing extension could be uncoupled, by systematically manipulating the level of activation using red-shifted optogenetic effectors. At low frequencies of photostimulation (≤20 Hz), a modest but significant increase in aggression was evoked, without any accompanying increase in unilateral wing extension behavior. Wing extension was evoked at photostimulation frequencies above 30 Hz. We note, however, that the ability to separate these two behaviors depended upon a number of experimental parameters, including light intensity, the type of effector used and the chromosomal insertion site of the effector. In general, however, weaker optogenetic stimulation tended to favor aggression, while higher levels favored wing extension. These data suggest that output from the P1 population can control different behaviors at different thresholds. However, we caution that the stimulation frequency-dependence of evoked aggression vs. wing extension behavior observed here may reflect our gain-of-function assay conditions (*Pan et al., 2012*), and may not necessarily apply during naturally occurring male-male or male-female social behavior. Nevertheless, it is of interest that studies of hypothalamic circuits in mice have also revealed a threshold-dependent influence of optogenetic activation on mating and aggression (although opposite to the direction observed here) (*Hong et al., 2014*; *Lee et al., 2014*). How and where these behavioral thresholds are set, and whether they involve common or distinct subsets of P1 neurons, will be an interesting topic for future investigation.

The temporal resolution afforded by optogenetic activation of P1 neurons also revealed an inverse relationship between the promotion of these two behaviors: at high stimulation frequencies (≥30 Hz), wing extension was promoted during the light ON phase, while aggression was suppressed; conversely, during the light OFF phase, aggression resumed and wing extension was suppressed. Our cellular epistasis experiment revealed that when aggression is inhibited, by silencing TK-GAL4[1] neurons (*Asahina et al., 2014*), wing extension triggered by optogenetic activation of P1[a] neurons persists into the light OFF phase, as described previously for single male flies

(*Inagaki et al., 2013*). This result indicates that aggressive behavior causally suppresses wing extension following the offset of photostimulation, suggestive of an inhibitory influence at the neural circuit or behavioral level. By extension, the suppression of lunging during the light ON phase may reflect a reciprocal inhibitory influence of courtship circuitry on aggression (however, the acute suppression of locomotor activity produced by strong optogenetic activation of P1 neurons may contribute to the inhibition of aggression as well). Rebound from such inhibition could then explain the rapid resumption of attack following the offset of high-intensity P1[a] photostimulation. The existence of reciprocal inhibitory influences between circuits controlling 'opponent' behaviors is well established (reviewed in *Kristan, 2008*). Our observations raise the possibility that such mutual antagonism may control the choice between courtship and aggression as well.

## P1 activation promotes a persistent internal state that facilitates future social behavior

Previously, it was shown that transient activation of P1 neurons triggers persistent wing extension in solitary males (*Inagaki et al., 2013*; *Bath et al., 2014*; *Clowney et al., 2015*). However, because that persistent effect was detected as ongoing singing, it was not possible to distinguish whether it reflected a persistent *internal* state, or rather reflected singing-dependent sensori-motor positive-feedback. Since aggression requires a physical interaction between two flies, we were able to distinguish these alternatives by isolating the flies from each other during and after optogenetic P1 activation. This experiment revealed that transient activation of P1 neurons could result in enhanced aggression, even when physical contact between the flies was delayed for 10 min, long after persistent singing had decayed to baseline. Thus, activation of P1 neurons can induce a persistent, fly-intrinsic internal state (e.g., of arousal or motivation) that facilitates future social behavior. This state can endure without any increase in locomotor activity, suggesting that it does not simply reflect a state of generalized arousal (*van Swinderen and Andretic, 2003*); however, it is difficult to prove that it exclusively influences social behavior.

The circuit-level mechanisms underlying the P1-induced persistent internal state remain to be elucidated. Potential mechanisms include the release of a long-lasting neuromodulator (*Flavell et al., 2013*), a recurrent excitatory circuit (*Wang, 2001*, *2008*), or a cell-autonomous mechanism (*Major and Tank, 2004*). Activation of P1 neurons themselves did not appear to trigger long-lasting persistent activity within this population, as measured by calcium imaging (Inagaki *et al.*, 2013 and *Figure 3—figure supplement 1*). Identification of P1 'follower cells' that display persistent activity will be an important step towards solving this problem.

## Interaction between P1 neurons and aggression circuits

Our results leave open the question of how P1 neurons promote aggression at the circuit level. In principle this influence could be direct or indirect. In a direct model, P1 neurons (or a specific subset thereof) could activate persistent aggression-specific circuits (*Figure 6A*). In an indirect model, P1 neurons would trigger a persistent internal state (*Figure 6B*), which is neutral with respect to aggression or courtship, and the influence of that state to promote aggression would be determined at a downstream site in the circuit. We observed that optogenetic activation of P1 neurons never triggered aggression in a stimulus-locked manner during the light ON phase, even during low frequency stimulation where no wing extension behavior was evoked; rather an increase in aggression was typically observed following PS offset (*Figure 3*, 10 Hz and 20 Hz). Such observations would seem to support an indirect model. However in preliminary studies, we have identified another small group of neurons from our screen, whose optogenetic activation promotes aggression but not courtship; these aggression-specific neurons can be activated by stimulation of P1 cells (Y. J., H. Chiu, E.D.H. and D.J.A., unpublished observations). Thus, it is possible that P1 neurons directly activate or prime persistent aggression-promoting circuitry, but that other effects of phasic photostimulation (e.g., locomotor arrest) interfere with overt fighting behavior.

How might such a multi-layered function play an adaptive role in social behavior? Male flies fight only in the presence of an appetitive resource, such as food and/or mating partners (*Jacobs, 1978*; *Hoffmann, 1987*; *Chen et al., 2002*; *Lim et al., 2014*; *Yuan et al., 2014*). Thus, aggression may occur when a male's brain detects both the presence of such a resource, and of a competitor male who threatens access to that resource. Since P1 neurons are activated by even brief contact with a

female (*Kohatsu et al., 2011*; *Clowney et al., 2015*; *Kohatsu and Yamamoto, 2015*), the persistent internal state they engender by this low-level activation could provide an enduring representation of this resource—i.e., a memory of a mating partner. We speculate that this representation could be combined at a later time and place with the detection of a competing male, thereby triggering aggression (*Figure 6C*). At the same time, the inhibitory influence of acute P1 activation on fighting may serve to suppress aggression towards a female when a male fly is directly engaged in courtship or mating behavior. In this way, P1 neurons could exert both inhibitory and facilitatory influences on aggression, albeit on different time scales.

## Implications for behavioral decision-making

The decision of whether to mate or fight is an important one for most animals, with implications for reproduction and survival at both the individual and species level. Our understanding of how this decision is made at the neural circuit level is limited. In principle, such a decision should reflect the combined influences of sensory cues (such as pheromones), past experience and internal state (*Certel et al., 2007*; *Certel et al., 2010*; *Palmer and Kristan, 2011*; *Palmer et al., 2014*). Tinbergen (*Tinbergen, 1951*) proposed that such decisions are made in a hierarchical manner. According to his view, this decision hierarchy is implemented in the nervous system as a series of feed-forward connections between circuit nodes that control progressively more specific aspects of behavior. Thus, for example, the decision to engage in predator defense vs. reproductive behavior is made before the decision to engage in a particular reproductive activity, such as mating vs. fighting; this decision is in turn made before the decision between different aggressive actions is taken (e.g., threat display vs. physical attack) (*Tinbergen, 1950*).

Although this model has relatively limited experimental support (*Pirger et al., 2014*), a clear prediction is the existence of a 'higher-level' circuit node (or module) that controls multiple reproductive behaviors, including aggression and courtship. Our discovery of a small cluster of neurons in *Drosophila* that collectively can control both fighting and mating behavior appears consistent with this prediction. Tinbergen also proposed that such common modules would be the site at which internal state influences exerted their effects to promote behavioral arousal or motivation (*Tinbergen, 1951*). Our finding that P1 neurons promote a persistent internal state that facilitates both courtship and aggression would seem consistent with this proposal as well. The independent discovery of a relatively small population of neurons that controls mating and fighting in the mouse hypothalamus (*Lin et al., 2011*; *Yang et al., 2013*; *Lee et al., 2014*), moreover, suggests that such common nodes may represent an evolutionarily conserved circuit 'motif' for the organization and control of innate social behaviors.

# Materials and methods

## Fly stocks

A list of the full genotypes for the flies used in each figure can be found in *Supplementary file 2*. Wild-type *Canton S* was originally obtained from M. Heisenberg (*Hoyer et al., 2008*). The following stocks were kindly provided: *w+;UAS-dTrpA1* (*Hamada et al., 2008*) was from P. Garrity (Brandeis University) and backcrossed into the *Canton S* background; *w+;PromE(800)-GAL4,tubP-GAL80ts*, *w+; UAS-Stinger* and *w+;UAS-hid,UAS-Stinger* stocks (*Billeter et al., 2009*) were from J. Levine; *dsx<sup>GAL4Δ2</sup>* (*Pan et al., 2011*) was from B. Baker (Janelia Research Campus); *5XUAS-eGFP::Kir2.1* (*Baines et al., 2001*; Bloomington #6595) and *20XUAS-IVS-CsChrimson::venus (attP2)* (*Klapoetke et al., 2014*; Bloomington #55136) and *13XLexAop2-CsChrimson::venus (attP40)* (Bloomington #55138) from V. Jayaraman (Janelia Research Campus). *TK-GAL4<sup>1</sup>* (*Asahina et al., 2014*) and *pJFRC2-10XUAS-ReaChR in attP40* (*Inagaki et al., 2013*) stocks were produced in the Anderson lab.

All Janelia GAL4 lines (*Pfeiffer et al., 2008*; *Jenett et al., 2012*) in the dTrpA1 activation screen (*Figure 1A and B*) were inserted in attP2. The parental GAL4 lines are available from Bloomington Stock Center: *R15A01-GAL4* (#48670), *R71G01-GAL4* (#39599), *R22G11-GAL4* (#49878) and *pBDPGAL4U* (referred to as "BDPG4U" in the main text and figures) (*Pfeiffer et al., 2010*). The LexA and split-GAL4 transgenic lines derived from the parental GAL4 lines, as well as *R57C10-AD* (attP40), *R57C10-DBD* (attP2), *pBPp65ADZpU* (attP40) and *pBPZpGAL4DBDU* (attP2) (referred to in

the text as BDP-AD and BDP-DBD, respectively) were obtained from G. Rubin and constructed as described in *Pfeiffer et al. (2010)*.

The *10XUAS-IVS-Syn21-Chrimson::tdT-3.1 in su(Hw)attP1* was generated (B. Pfeiffer, personal communication) by cloning Chrimson (*Klapoetke et al., 2014*) to include a C-terminal fusion of a *Drosophila* codon-optimized tdTomato fluorescent tag in a 10XUAS vector (*Pfeiffer et al., 2010*) including a Syn21 leader (*Pfeiffer et al., 2012*), ER export signal (FCEYENEV), and membrane-trafficking signal from Kir2.1 (*Gradinaru et al., 2010*). The *20XUAS-IVS-Syn21-OpGCaMP6s-p10 in su (Hw)attP5* was generated by cloning a *Drosophila* codon-optimized GCaMP6s (B. Pfeiffer personal communication; *Chen et al., 2013*) to include a Syn21 translation leader sequence in pJFRC82-20XUAS-IVS-Syn21-GFP-p10 after removal of GFP (*Pfeiffer et al., 2012*).

## Thermogenetic activation

All aggression assays were performed in acrylic multi-chamber aggression arenas described in *Asahina et al. (2014)*. Each arena contained 12 cylindrical chambers in a 4x3 array that measured 12 mm high x 16 mm diameter. The walls of the chamber were coated with Insect-a-Slip (Bioquip Products, Rancho Dominguez, CA) and the clear acrylic top plate was coated with SurfaSil Siliconizing Fluid (Thermo Fisher Scientific, Waltham, MA) to discourage flies from walking on these surfaces. The floor of the arenas was composed of clear acrylic covered with a uniform layer (~1 mm thick) of apple juice-agarose food (2.5% (w/v) sucrose and 2.25% (w/v) agarose in apple juice). The arenas were illuminated from beneath with visible light by Cold Cathode fluorescent backlights (Edmund Optics, Barrington, NJ) and ambient overhead room lighting. Flies were introduced into the chambers by gentle aspiration through a hole in the top plate and allowed to acclimate to the chamber for 2–5 min prior to the start of the recording. In all experiments, males from separate vials were paired in the same chamber during the assay, so the males were naïve to each other.

Males were reared at 21°C, collected 1–2 days after eclosion and group-housed at a density of 16–18 males per vial at 21°C on standard fly medium for 6–7 days. Rearing and housing were done in a light cycling incubator set for a 16/12 hr light/dark cycle. dTrpA1 activation experiments were run at 29°C, with the exception of experiments shown in *Figure 2* and *Figure 1—figure supplement 3*, which were run at 32°C to increase efficacy of dTrpA1 activation. Low temperature controls were run at 21°C.

To differentiate males in the mixed-genotype pair experiments (*Figures 1G,H* and *Figure 5—figure supplement 1A,B*) we clipped the tip of one male's wing. This identifying feature was used to manually correct the fly identities after automated tracking (see below). Wing clipping was done two days prior to the behavior assay by cold-anesthetizing the males on ice. A similar surgical procedure was used for wingless male experiments.

## Optogenetic activation

A detailed description of the setup used for optogenetic activation experiments can be found in *Inagaki et al. (2013)*. The assays were performed essentially as described for thermogenetic activation with the following differences. Experiments were performed in 8-well arenas (2x4 array) with IR backlighting (855 nm, SOBL-200x150-850, SmartVision Lights, Muskegon, MI). A high-powered LED was mounted 11 cm above the top of each chamber at a 24° angle to maximize light exposure. Photostimulation (PS) with different wavelengths of light was performed with the following LEDs: Rebel 10 mm square CoolBase LEDs (Luxeon Star LEDs, luxeonstar.com) at 530 nm (125 lm at 700 mA, SR-05-M0070) and 627 nm (102 lm at 700 mA, SR-05-D2050), and 680-690 nm 1w Red LED with star PCB (FD-14R-Y1, www.ledfedy.com). Light intensity was measured with a photodiode power sensor (S130VC, Thorlabs, Newton, NJ) placed at the location of the behavior chambers with the arena absent. Measurements were taken at different voltage inputs and averaged across four readings.

For both ReaChR and CsChrimson experiments, males were raised at 25°C on standard fly media, collected 1−2 days after eclosion and housed in the dark for 6−7 days on food containing 0.4 mM all *trans*-Retinal (Sigma-Aldrich, St. Louis, MO). The specific stimulation protocols for each experiment are described in the figure legends.

Sliding door experiments (*Figure 5*) were performed in behavior arenas identical to those used in optogenetic experiments with the exception that a vertical metal barrier divided each chamber in half. Males were loaded on separate sides of the barrier and allowed to acclimate for 2 min before

recording. The barriers were manually removed after the end of the PS and the time period from 1 s prior to 10 s after barrier removal was not tracked (see below), because the movement of the barrier interfered with tracking.

Ablation of oenocytes (*Figure 3I*) was carried out as described in *Wang et al. (2011)*. Males were heatshocked for 6 days prior to the behavior assay with 12 hr at 29°C and 12 hr recovery at 25°C. Males were placed at 25°C to recover from heatshock for 12 hr prior to assay.

## Fly tracking and behavior classification

Male-male interactions were recorded at 30 frames per second using gVision (http://gvision-hhmi. sourceforge.net) video acquisition software run with Matlab (Mathworks, Natick, MA). For the GAL4 neuronal activation screen, pairs of males were tracked and automated scores of lunging and unilateral wing extension behaviors were derived using CADABRA software (*Dankert et al., 2009*). For all other experiments shown in the paper, flies were tracked using Caltech FlyTracker software (Eyrún Eyjólfsdóttir & Pietro Perona, Caltech), which is available for download at http://www.vision.caltech. edu/Tools/FlyTracker/. For mixed-genotype experiments, the identities of the fly tracks were manually corrected using the visualizer.m GUI that is available with the FlyTracker package.

Analysis of lunging and wing extension behaviors in all experiments except the GAL4 screen was performed with classifiers developed using JAABA software (*Kabra et al., 2013*). For a description of the video datasets that were used to train the classifiers see *Supplementary file 3A*. Performance of the classifiers against an independent set of manually scored videos is described in *Supplementary file 3B*. To validate classifiers we manually scored a set of ground-truth videos for lunging and wing extension using the 'ground-truthing' mode in JAABA. A randomly selected set of frames from the ground-truth videos was manually labeled as positive or negative for the behavior of interest on a frame-by-frame basis. Frames where there was no uncertainty as to behavior label were labeled as 'important' to distinguish them from frames where we could not unambiguously decide the label. The error rates, precision and recall between the ground-truth and predicted labels were calculated for all frames labeled, as well as for the 'important' frames only (see *Supplementary file 3*). For experiments where a small difference in wing extensions between experimental and control genotypes was observed (*Figures 2*, *3* and *4D−F*), the wing extension scores were manually curated to eliminate possible false-positives. Wing extensions of the separated males in *Figure 5* were manually scored.

## Statistical analyses

Statistical analyses were performed using Matlab and Prism6 (GraphPad Software, La Jolla, CA). All behavioral data were analyzed with nonparametric tests. The cutoff for significance was set as an alpha $\alpha<0.05$. The number of pairs tested in each experiment is specified in the figure legends. Sample size was determined based on an initial power analysis performed on our parental GAL4 screen. Each experiment was repeated at least twice on independent groups of flies. Outliers were defined as data points falling outside 1.5-times the interquartile range of the data, and were not excluded from statistical analyses. For pairwise *between* group comparisons we performed Mann-Whitney *U*-tests, and for pairwise *within* group comparisons we used Wilcoxon signed-rank tests. Comparisons of three or more groups that did not involve repeated measures were performed using the Kruskal-Wallis (ANOVA) test. In the case of a significant main effect, post-hoc pairwise Mann-Whitney *U*-tests were performed. For within genotype comparisons over multiple treatments we performed a Friedman's test. In the case of a significant main effect, post-hoc pairwise Wilcoxon signed-rank tests were performed. In both cases, post-hoc test were corrected for multiple comparisons using Dunn's or Bonferroni correction, as noted in the figure legends. A sign test was used to determine whether the proportion of lunging to wing extensions at different light frequencies (*Figure 3G*) was different from a hypothetical median value of zero.

Identification of behavioral hits in neuronal activation screen was done by testing for GAL4 lines that showed a statistical increase in lunging, tussling, wing threat or wing extension with dTrpA1 at 29°C compared to pBDPGAL4U controls, which were pooled from the entire screen (n=226 pairs). A Kruskal-Wallis test was used, followed by Mann-Whitney *U*-tests between the pBDPGAL4U control and individual lines, which were corrected for multiple comparisons using the Benjamini-Hochberg

procedure. These hits were then verified by manual inspection of behavior scores to eliminate false-positives from the CADABRA behavior classifiers.

Linear regressions (*Figures 3E and F* and *Figure 3—figure supplement 1G*) and non-linear regressions (*Figure 3—figure supplement 2E* and *Figure 5—figure supplement 1E*) were done in Prism6. The fraction of pairs performing lunging or wing extension was binned in 10 s time intervals. A linear model was fit to the data and the best-fit value of the slope was calculated. For non-linear regressions, a sigmoidal function was fit to the data and the best-fit value for the half-maximal response ($t_{1/2}$) was calculated. Goodness of fit was tested by two-way ANOVA between the predicted function (line or sigmoid) and the actual fraction of flies performing the behavior, which indicated a good fit in all cases.

## Fly histology

Dissections, fixation and staining of adult brains and ventral nerve cords were carried out at 4°C unless otherwise specified. Brains were dissected in PBS and fixed in 4% paraformaldehyde in PBS for 1 h, incubated overnight in blocking solution of 5% normal goat serum in PBT (PBS + 0.05% Triton X-100), followed by a 2–3 day incubation in primary antibody in blocking solution and 2–3 days in secondary antibody in blocking solution. Brains were washed between each incubation step with six 15 min washes in PBT. After staining, brains were incubated overnight in Vectashield (Vector Laboratories, Burlingame, CA) before mounting. Confocal serial optical sections were collected either with a Zeiss LSM710 confocal microscope (Zeiss, Thornwood, NY) with a Plan-Apochromat 20x/0.8 M27objective and a Plan-Apochromat 63x/1.4 oil immersion objective, or a FluoView FV1000 Confocal laser scanning microscope (Olympus, Waltham, MA) with a 30×/1.05-NA silicone oil objective (Olympus). Image preparation and adjustment of brightness and contrast were done in Fiji (http://fiji.sc/), and unless otherwise noted are maximum Z-projections of the confocal stacks. Brains and VNCs were imaged separately and arranged into a composite images using Photoshop (Adobe).

For primary antibodies we used rat antibody to GFP (A11122, 1:500; Thermo Fisher Scientific), mouse antibody to Bruchpilot (nc82, 1:50, Developmental Studies Hybridoma Bank [DSHB], University of Iowa, Iowa City, IA), mouse antibody to myc (9E10, 1:50, DSHB), rabbit antibody to Fru$^{M}$ (1:3000, *Stockinger et al., 2005*) and chicken antibody to GFP (ab13970, 1:2000, abcam, Cambridge, MA). For secondary antibodies we used Alexa Fluor 488 (1:500, goat anti-rabbit, goat anti-mouse, Thermo Fisher Scientific), Alexa Fluor 568 (1:500, goat anti-rabbit, goat anti-mouse, Thermo Fisher Scientific) and Alexa Fluor 633 (1:500, goat anti-mouse, Thermo Fisher Scientific).

## In vivo calcium imaging

Males for imaging experiment were reared and housed using the same protocol as the optogenetic experiments. Brains were imaged in an intact male using a head fixed preparation as described in *Inagaki et al., (2012)*. The flies were anesthetized on ice and mounted on the underside of a thin plastic plate with a well for *Drosophila* imaging saline (108 mM NaCl, 5 mM KCl, 2 mM CaCl$_2$, 8.2 mM MgCl$_2$, 4 mM NaHCO$_3$, 1 mM NaH$_2$PO$_4$, 5 mM trehalose, 10 mM sucrose, 5 mM HEPES, pH 7.5) such that the fly head was immersed in the solution. A small window in the cuticle was made and the fat bodies and air sacs were gently removed to give a clear view of the brain. The saline was changed prior to imaging to remove any debris.

Two-photon imaging was performed essentially as described in *Inagaki et al. (2013)*. Imaging was performed on an Ultima two-photon laser-scanning microscope (Bruker) with an imaging wavelength of 940 nm. We used a 525/50 nm (center wavelength/bandwidth) band-pass filter (ET525/50m, Chroma Technology, Bellow Falls, VT) in the emission pathway to detect the GCaMP6s fluorescence. The scanning resolution was 128×128 pixels with an 8 μsec dwell time per pixel, 4x optical zoom and 1 Hz scanning speed. A 40×/0.80-NA water-immersion objective (Olympus) was used for imaging.

To activate Chrimson we used a high-power deep-red LED (660 nm) collimated with an optic fiber (M660F1, Thorlabs). The optic fiber was connected to a fiber optic holder (Product #56200, World Precision Instruments, Sarasota, FL) equipped with a 660 nm band-pass filter (Product #67836, Edmund Optics) to narrow the bandwidth of the LED output, and a neutral density filter (OD2.0; NE20B-A, Thorlabs) to reduce the light intensity. We used a 200 μm core multimode optic fiber (NA,

0.39; FT200EMT, Thorlabs) to deliver light from the fiber optic holder to the brain. A bare tip was custom-made on one side of the fiber and this tip was immersed in the saline bath and placed 250 µm away from the brain.

We set the light intensity to be 0.93 µW at the tip of the optic fiber. At a distance of 250 µm from the tip of a 0.39-NA optic fiber the light power at the surface of the brain was calculated to be approximately 27.5 µW/mm$^2$. This light intensity was determined empirically by optimizing the neutral density filter and input voltage to the LED to find a condition that gave robust GCaMP responses in P1 neurons at the maximum PS frequency (50 Hz) and no responses at minimum PS frequency (1 Hz). A 10 ms pulse-width was used at each frequency.

Images were collected from multiple focal planes to ensure that the majority of P1 cells were captured. The PS protocol (see *Figure 3—figure supplement 1B*) was delivered at each focal plane before imaging the next. An ROI was manually drawn around each cell body and the fluorescence signal from the ROI was smoothed with a moving average (window = 5 frames). The average signal before stimulation was used as $F_0$ to calculate the $\Delta F/F_0$ ($\Delta F/F$). For each cell, a single focal plane in which we observed the highest $\Delta F/F$ was used for analysis. Normalized $\Delta F/F$ values for each cell were calculated by dividing $\Delta F/F$ by the maximum $\Delta F/F$ of the cell. The peak $\Delta F/F$ at each stimulation frequency was selected by finding the maximum normalized $\Delta F/F$ during the 30 s stimulation window. For the pre-stimulation peak value, we determined the maximum during the 30 s window 5 s prior to the first PS.

## Acknowledgements

We thank Barret Pfeiffer for design, construction, and gift of the Chrimson and OpGcamp lines, construction of the GAL4 collection and LexAop-Flp lines and for advice and discussions; Hui Chiu for contributions, discussions and for sharing her unpublished data; Heather Dionne and Christine Murphy for construction of split-GAL4 and LexA lines; Kenta Asahina, Stefanie Hampel, Andrew Seeds, Moriel Zelikowsky, Weizhe Hong and Allan Wong for advice and discussions; Richard Axel for critical comments on the manuscript; Pietro Perona and Eyrún Eyjólfsdótir for FlyTracker software; Kristin Branson and Mayank Kabra for help with JAABA classifiers; Frank Midgley and Jinyang Liu for help modifying the CADABRA software for a computer cluster; Guss Lott for gVision software; Barry Dickson, Bruce Baker and Yufeng Pan for GAL4 lines; Sharon Low and Mary Phillips for technical assistance with behavioral assays; Karen Hibbard, Monti Mercer, James McMahon and Marcela Arenas-Sanchez for fly husbandry; Celine Chiu for laboratory management; Gina Mancuso, Crystal Sullivan, Sarah Moorehead and Emily Willis for administrative assistance; Kevin Moses, the Janelia Visiting Scientist Program and the Fly Olympiad Program for support.

## Additional information

### Funding

| Funder | Grant reference number | Author |
|---|---|---|
| National Institutes of Health | Ruth L. Kirschstein National Research Service Award (NRSA) | Eric D Hoopfer |
| Heiwa Nakajima Foundation | | Hidehiko K Inagaki |
| Howard Hughes Medical Institute | | Gerald M Rubin David J Anderson |
| National Institutes of Health | DA031389 | David J Anderson |
| Gordon and Betty Moore Foundation | | David J Anderson |

The funders had no role in study design, data collection and interpretation, or the decision to submit the work for publication.

## Author contributions

EDH, Conducted all experiments except for the functional imaging, Conception and design, Acquisition of data, Analysis and interpretation of data, Drafting or revising the article; YJ, Conducted the functional imaging experiment, Acquisition of data, Analysis and interpretation of data; HKI, Generated the ReaChR transgenic flies and established the optogenetic behavior system, Drafting or revising the article, Contributed unpublished essential data or reagents; GMR, Contributed to the design of genetic intersection strategies, Conception and design, Contributed unpublished essential data or reagents; DJA, Conception and design, Analysis and interpretation of data, Drafting or revising the article

# Additional files

### Supplementary files

• Supplementary file 1. Genetic intersections used to target P1 neurons.

• Supplementary file 2. Full genotypes of flies in each experiment.

• Supplementary file 3. Training datasets and performance of JAABA classifiers. (A) The videos and genotypes of flies used to train the lunging and wing extension classifiers. The total number of bouts of each behavior, along with the total number of frames scored as positive or negative for the behavior are shown. (B) Performance of the classifiers against an independent set of 'ground-truth' videos. The framewise error rates and performance of the classifiers was calculated from a set of between 4,000 and 32,000 frames of manually scored ground-truth labels (total frames and total bouts). Frames were labeled as 'Important' if there was no uncertainty about the behavioral label to distinguish them from frames where the human scorer was unsure of the label. For each classifier the error rates, precision and recall for 'important' vs. all frames is shown. See Materials and methods for detailed description of ground-truth validation.

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
