## [Decision Letter]

Thank you for submitting your work entitled "P1 interneurons promote a persistent internal state that enhances male aggression as well as courtship in *Drosophila*" for consideration by *eLife*. Your article has been reviewed by two peer reviewers and the evaluation has been overseen by Mani Ramaswami as Reviewing Editor and a Senior Editor.

The reviewers have extensively discussed the reviews with one another and the Reviewing editor has drafted this decision to help you prepare a revised submission.

Summary:

This manuscript provides compelling evidence that P1 interneurons can promote inter-male aggression and thus are not solely involved in courtship behavior, as generally assumed. The core claim(s) presented by these authors is that P1 neurons (at least a subset of them) mediate aggression as well as courtship. This is based on findings of an innovative and impressive screen for neurons whose activation triggers aggressive behaviors, as well as careful genetic follow up experiments, that identify a role for P1 neurons in aggression as well as the already established role in courtship. The claim is extended to the idea that P1 neurons are multiplex units as illustrated by differential functional outcomes associated with differential levels of stimulation. Deploying optogenetic stimulation in different ways, the authors show that weak activation of P1 neurons promotes aggression and no courtship, while stronger activation evokes wing extension, i.e. courtship behavior, and suppressed aggression. Hence both social behaviors are reciprocally controlled by activation of P1 neurons representing an antagonistic interaction between the aggression and courtship circuitry. In addition the authors show that aggression behavior can persist more than 10 min. and requires chemosensory cues on a conspecific target to be induced.

The manuscript stands out for the excellence of the experiments, the quality of the data, and for showing for the first time that central P1 neurons have potential functions in behaviors outside courtship, and thus enabling new experiments to dissect the diversity of P1 neurons as well as their potentially diverse upstream and downstream circuitry.

Overall criticisms to be addressed in a rewritten manuscript:

A major issue with the paper is that many of the most interesting issues raised by the authors are actually inferences consistent with the experiments but not proven or even tested by the experiments performed. There is disappointment when problems and questions raised in the Introduction are not actually resolved. Therefore, the authors should rewrite the Introduction, and where appropriate the lead-in to experimental sections with a clear focus on what is actually shown by the data, and consider potential wider implications of the observations to a revised Discussion. A revised manuscript should explicitly acknowledge (or address through decisive new experiments) questions that remain to be addressed at the end of this work.

Below are examples of issues that are not resolved.

A) We do not know that P1 neurons actually respond (in situ) to differentially to ambient stimuli (pheromones, behavioral displays, attacks, sexual encounters). Thus, we see how a decision might be made but there is nothing that directly speaks to how decisions are actually made, let alone whether they are made as shared hierarchically organized modules or labeled lines.

B) We do not know whether different subpopulations of P1 neurons are involved. We do not even know which P1 neurons are in play here (recent estimates in Pan and Baker state that there are ~40 of these and most experiments in study is work with ~10).

C) Does low vs high stimulation lead to different levels of activity for each neuron or recruitment of multiple neurons? Does it refer to different stimulus threshold for different neuronal subpopulations?

D) Are P1 neurons involved as determinants of male-male courtship and male-male aggression as well as male-female courtship? If they are, is the decision to do one or the other of these things a consequence of levels of stimulation or is it downstream? It seems plausible that specific subpopulations of P1 neurons determine the level and perhaps quality of arousal and do not specifically determine specific outcomes. On the other hand to the extent that circuitry down stream of P1 neurons is known, it is usually discussed with regard to effectors. With this in mind is the downstream neural circuitry that drives male-male courtship very different from aggression?

E) What does the decision look like at a neural or biochemical level?

Essential revisions:

1) The question re labeled lines or overlapping circuit nodes (Introduction, second paragraph) that places the work in a context established by Anderson and Tinbergen does not belong in the Introduction. Presenting these questions and issues here leads to the expectation that the authors have resolved these issues in the paper. However, as implicitly acknowledged in the Discussion, these are not resolved or addressed by the experiments presented. Anderson and Tinbergen could be more appropriately considered in a revised Discussion. Which should also consider a parsimonious explanation consistent with the authors' observations, that P1 interneurons promote state or arousal that may not be restricted to aggression and mating. For all we know P1 neurons may feed into additional behavioral options that are context dependent.

2) In the last paragraph of the Introduction the authors state a broad hypothesis clearly that is not resolved by the data. The data show that when the neurons are manipulated these responses can be observed. There is no evidence (as the authors later acknowledge in the Discussion) that P1 neurons are active for aggression under normal conditions.

3) The absence of evidence that this aggression is via social feedback does not prove that the signals are "fly-autonomous."

4) The second paragraph of the subsection “A large-scale neuronal activation screen yields multiple independent hits promoting both male courtship and aggressive behavior” claims that these findings are interesting because of the relationship between aggression circuitry and courtship circuitry in mammals. This relationship has been linked to gene pathways in flies by the Kravitz lab in de la Paz Fernandez et al. (2010) and by other work from David's lab – both cited later.

In the second paragraph of the subsection “P1 neuron activation promotes inter-male aggression” of the Discussion, the authors summarize aspects of the thorny issue being raised in this critique. They also point to some specific missing lines of proof that would be required to justify the claims that appear to be made in this manuscript.

---

## [Author Response]

Overall criticisms to be addressed in a rewritten manuscript:

*A major issue with the paper is that many of the most interesting issues raised by the authors are actually inferences consistent with the experiments but not proven or even tested by the experiments performed. There is disappointment when problems and questions raised in the Introduction are not actually resolved. Therefore, the authors should rewrite the Introduction, and where appropriate the lead-in to experimental sections with a clear focus on what is actually shown by the data, and consider potential wider implications of the observations to a revised Discussion. A revised manuscript should explicitly acknowledge (or address through decisive new experiments) questions that remain to be addressed at the end of this work.* We agree with the reviewers’ comment. To address this point, we have completely rewritten the Introduction and Discussion, as well as certain key transition sections in the Results. As requested, all mention of behavioral decision-making, Tinbergen, etc. has now been removed from the Introduction and shifted to the final section of the Discussion. We now frame the study in a way that emphasizes its central objective and novel findings, and separates them clearly from the important but unresolved issues raised by the reviewers. These latter are now addressed in the new Discussion, in different subsections.

*Below are examples of issues that are not resolved.*

*A) We do not know that P1 neurons actually respond (in situ) to differentially to ambient stimuli (pheromones, behavioral displays, attacks, sexual encounters). Thus, we see how a decision might be made but there is nothing that directly speaks to how decisions are actually made, let alone whether they are made as shared hierarchically organized modules or labeled lines.*

Studies from other laboratories, cited in the original and current versions, show that P1 neurons respond to pheromonal as well as to visual cues. We agree that our paper does not show how the decision to mate vs. fight is actually made, and apologize if we gave the impression that we intended to solve and had solved this problem; it was presented as the broader context to which this work is relevant. We have now de-emphasized this point until the end of the Discussion, in the revised version.

*B) We do not know whether different subpopulations of P1 neurons are involved. We do not even know which P1 neurons are in play here (recent estimates in Pan and Baker state that there are ~40 of these and most experiments in study is work with ~10).*

We agree that this is an important and unresolved issue. The neurons manipulated in our P1^a^ split GAL4 line are a deterministic subset of the ~40 neurons identified by Pan and Baker. Our results from the *fru^FLP^* intersection experiments (Figure 2) show that activation of as few as 2-5 P1 neurons/hemibrain can produce aggression but not courtship. However it is not possible to conclude whether this is a deterministic (aggression-specific) or stochastic subset of the cells. The frequency titration experiments using CsChrimson (Figure 3), and the new imaging data supporting these experiments in Figure 3—figure supplement 1 (see point (C), below) are consistent with the idea that the average level of activity among the P1^a^ subset of P1 neurons determines the behavioral outcome. However, they do not preclude the possibility of functional heterogeneity within this subpopulation, as we acknowledged in the original submission. Our ability to resolve this issue at the level of single, identified neurons is currently limited by available technology. We have acknowledged this as an open question in the revised Discussion: “How and where these behavioral thresholds are set, and whether they involve common or distinct subsets of P1 neurons, will be an interesting topic for future investigation”.

*C) Does low vs high stimulation lead to different levels of activity for each neuron or recruitment of multiple neurons? Does it refer to different stimulus threshold for different neuronal subpopulations?*

This is an excellent question. To address this issue we have added a new supplemental Figure 3—figure supplement 1 that shows the neuronal responses of P1^a^ neurons to increasing frequencies of Chrimson stimulation. Briefly, we co-expressed GCaMP6s and Chrimson with P1^a^ spGAL4 and stimulated at the same frequencies used in the behavioral experiments of Figure 3. We find that the 6 P1^a^ neurons sampled (in each of n=3 flies) show a roughly linear increase in activity across the photostimulation range of ~10-50Hz, and that the normalized response is similar across the sampled neurons. This suggests that increasing the stimulation frequency increases the activity of each P1^a^ neuron to a similar extent, and provides no evidence of differential recruitment of different subsets at different stimulation thresholds.

*D) Are P1 neurons involved as determinants of male-male courtship and male-male aggression as well as male-female courtship? If they are, is the decision to do one or the other of these things a consequence of levels of stimulation or is it downstream? It seems plausible that specific subpopulations of P1 neurons determine the level and perhaps quality of arousal and do not specifically determine specific outcomes. On the other hand to the extent that circuitry down stream of P1 neurons is known, it is usually discussed with regard to effectors. With this in mind is the downstream neural circuitry that drives male-male courtship very different from aggression?*

All of the reviewers’ questions are on point and are considered in the revised Discussion (subsection “Interaction between P1 neurons and aggression circuits”). We have now included two new schematics in Figure 6 to illustrate two possible models to explain the results. To answer the reviewers’ last question, the immediate downstream neural circuitry that drives male-male courtship involves a descending interneuron called pIP10. Activation of pIP10 neurons promoted wing extension (as reported by von Philipsborn et al. (2011), but in our hands did not promote aggression. In preliminary experiments, we have identified a small group (2-3/hemibrain) of cells that are activated by P1 neurons; optogenetic activation of these neurons in freely behaving flies promotes aggression but not courtship. These unpublished data are now mentioned in the aforementioned subsection. Together these results suggest that courtship and aggression circuits likely diverge downstream of P1 neurons.

*E) What does the decision look like at a neural or biochemical level?*

The reviewer appears to be asking about the molecular and circuit-level mechanisms that control the decision between aggression vs. courtship. We share the reviewers’ curiosity. Our results identify an important cellular point-of-entry to this problem, by revealing an unexpected connection between P1 neurons and aggression. However, elucidating the biochemical or circuit mechanisms that underlie this decision will take years of further work, and is beyond the scope of the present study.

*Essential revisions:*

*1) The question re labeled lines or overlapping circuit nodes (Introduction, second paragraph) that places the work in a context established by Anderson and Tinbergen does not belong in the Introduction. Presenting these questions and issues here leads to the expectation that the authors have resolved these issues in the paper. However, as implicitly acknowledged in the Discussion, these are not resolved or addressed by the experiments presented. Anderson and Tinbergen could be more appropriately considered in a revised Discussion. Which should also consider a parsimonious explanation consistent with the authors' observations, that P1 interneurons promote state or arousal that may not be restricted to aggression and mating. For all we know P1 neurons may feed into additional behavioral options that are context dependent.* We take the reviewers' point. As mentioned above, we have now moved the discussion of the potential relevance of our observations to Tinbergen’s model from the Introduction to the Discussion section (subsection “Implications for behavioral decision-making”). This discussion is now clearly presented as a speculative inference at the end of the paper, rather than as a central question raised and answered by the study.

The review also raised the question of whether the state or arousal promoted by P1 neurons is specific to aggression and mating, or whether it might be more general and affect other behaviors. Since it is not practical to explore the entire space of fly behaviors that might be affected by manipulating P1 cells, this is a difficult question to answer conclusively, as we now acknowledge in the Discussion (subsection “P1 activation promotes a persistent internal state that facilitates future social behavior”). However, our data do show that transient P1 activation does not cause any persistent increase in locomotor activity in solitary flies (see Figure 3—figure supplement 2 and Figure 5—figure supplement 1). As increased locomotion is considered a sine qua non of enhanced behavioral arousal in flies, this provides some evidence against the "parsimonious explanation" advanced by the reviewers. To what extent P1 neurons influence other non-social behaviors remains an interesting question for future investigation.

*2) In the last paragraph of the Introduction the authors state a broad hypothesis clearly that is not resolved by the data. The data show that when the neurons are manipulated these responses can be observed. There is no evidence (as the authors later acknowledge in the Discussion) that P1 neurons are active for aggression under normal conditions.*

We agree that this is an important caveat of our work. Unfortunately it is currently not technically possible to monitor the activity of neurons during aggression under normal conditions in freely behaving flies. We now point this out and make explicit this caveat, in the subsection “P1 neurons can promote aggression as well as courtship” of the revised Discussion. We have also revised the wording of the Introduction to emphasize that our results indicate what P1 cells *can* do (lines 96-99).

*3) The absence of evidence that this aggression is via social feedback does not prove that the signals are "fly-autonomous."*

We have changed the wording to "fly-intrinsic" in the sense that it does not require social behavioral signals extrinsic to the fly. We have also clarified that the motivation for the mixed-pair experiment is to address the distinction between an increase in aggression that depends on a reciprocal exchange of social behavioral signals between the two males, vs. an increase that is intrinsic to one male and does not depend on counter-attack or courtship from the other male (subsection “P1-enhanced aggressiveness is fly-intrinsic and not a social response to increased courtship”, first paragraph).

*4) The second paragraph of the subsection “A large-scale neuronal activation screen yields multiple independent hits promoting both male courtship and aggressive behavior” claims that these findings are interesting because of the relationship between aggression circuitry and courtship circuitry in mammals. This relationship has been linked to gene pathways in flies by the Kravitz lab in de la Paz Fernandez et al. (2010) and by other work from David's lab – both cited later.*

The context of this transition was the link between courtship and aggression circuits. De la Paz Fernandez et al. (2010) focused on chemosensory signals, not circuits. We have now referred in this context to other work from the Kravitz lab that links the control of aggression and courtship behavior to octopaminergic neurons (subsection “A large-scale neuronal activation screen yields multiple independent hits promoting both male courtship and aggressive behavior“, second paragraph). We have also deleted the statement that these findings are "interesting," and stated simply that they "drew our attention".

*In the second paragraph of the subsection “P1 neuron activation promotes inter-male aggression” of the Discussion, the authors summarize aspects of the thorny issue being raised in this critique. They also point to some specific missing lines of proof that would be required to justify the claims that appear to be made in this manuscript.*

The reviewers note that we pointed out the caveats of interpretation in our original submission. Since the comment above is a statement of fact and not a question, it is not clear what additional revision is being requested. We have revised this section in the Discussion, and have now explicitly stated that "Thus, while activation of P1 neurons can enhance aggression, it remains unclear whether they are normally required for this behavior, and if so under what circumstances."